# High-Density Lipoprotein (HDL) Inhibits Serum Amyloid A (SAA)-Induced Vascular and Renal Dysfunctions in Apolipoprotein E-Deficient Mice

**DOI:** 10.3390/ijms21041316

**Published:** 2020-02-15

**Authors:** Xiaoping Cai, Gulfam Ahmad, Farjaneh Hossain, Yuyang Liu, XiaoSuo Wang, Joanne Dennis, Ben Freedman, Paul K. Witting

**Affiliations:** 1Discipline of Pathology, Charles Perkins Centre, Faculty of Medicine and Health, The University of Sydney, Sydney, NSW 2006, Australia; xcai4874@uni.sydney.edu.au (X.C.); gulfam.ahmad@sydney.edu.au (G.A.); farjaneh@hotmail.com (F.H.); anna.liu@sydney.edu.au (Y.L.); xiaosuo.wang@sydney.edu.au (X.W.); jo-dennis@optusnet.com.au (J.D.); 2Heart Research Institute, Charles Perkins Centre, and ANZAC Research University of Sydney, Sydney, NSW 2006, Australia; ben.freedman@sydney.edu.au

**Keywords:** renal inflammation, atherosclerosis, acute phase protein, oxidative damage

## Abstract

Serum amyloid A (SAA) promotes endothelial inflammation and dysfunction that is associated with cardiovascular disease and renal pathologies. SAA is an apoprotein for high-density lipoprotein (HDL) and its sequestration to HDL diminishes SAA bioactivity. Herein we investigated the effect of co-supplementing HDL on SAA-mediated changes to vascular and renal function in apolipoprotein E-deficient (ApoE^−^/^−^) mice in the absence of a high-fat diet. Male ApoE^−/−^ mice received recombinant human SAA or vehicle (control) by intraperitoneal (i.p.) injection every three days for two weeks with or without freshly isolated human HDL supplemented by intravenous (i.v.) injection in the two weeks preceding SAA stimulation. Aorta and kidney were harvested 4 or 18 weeks after commencement of treatment. At 4 weeks after commencement of treatment, SAA increased aortic vascular cell adhesion molecule (VCAM)-1 expression and F_2_-isoprostane level and decreased cyclic guanosine monophosphate (cGMP), consistent with SAA stimulating endothelial dysfunction and promoting atherosclerosis. SAA also stimulated renal injury and inflammation that manifested as increased urinary protein, kidney injury molecule (KIM)-1, and renal tissue cytokine/chemokine levels as well as increased protein tyrosine chlorination and P38 MAPkinase activation and decreased in Bowman’s space, confirming that SAA elicited a pro-inflammatory phenotype in the kidney. At 18 weeks, vascular lesions increased significantly in the cohort of ApoE^−/−^ mice treated with SAA alone. By contrast, pretreatment of mice with HDL decreased SAA pro-inflammatory activity, inhibited SAA enhancement of aortic lesion size and renal function, and prevented changes to glomerular Bowman’s space. Taken together, these data indicate that supplemented HDL reduces SAA-mediated endothelial and renal dysfunction in an atherosclerosis-prone mouse model.

## 1. Introduction

Endothelial dysfunction is a hallmark of atherosclerosis and represents an early event in developing cardiovascular disease (CVD). Endothelial cells function in a multifaceted manner to regulate and modulate vascular tone and mediate immune and inflammatory responses. Perturbations of endothelial cell structure and function in acute and chronic inflammatory diseases can promote endothelial dysfunction, which is an early pathological process causally linked to the development of atherosclerosis [1,2], a process central to developing CVD.

Serum amyloid A (SAA) is an acute phase protein and inflammation marker. Levels of circulating SAA increase ~1000-fold in response to microbial infection or trauma, and persistently high levels are evident in chronic pathologies such as diabetes mellitus [3], rheumatic disorders [4], various cancers [5,6] and chronic inflammatory disorders such as atherosclerosis [7]. The hepatocyte is the primary source of SAA and these cells yield the acute phase protein in response to inflammatory cytokines, including tumor necrosis factor (TNF), and interleukin 1 and 6 (IL-1 and IL-6) [8]. In addition, stimulated monocytes/macrophages, vascular smooth muscle cells, and endothelial cells synthesize SAA. Deposits of the acute phase protein are detected in human plaque [9,10]. Whether SAA in the vascular wall elicits inflammatory responses is presently unclear. However, the potential for SAA to instigate endothelial [11,12,13] and renal dysfunction has been established [14]. Taken together, these data suggest that SAA is not merely a biomarker of inflammation but that it also stimulates cells through binding various surface receptors [15] to elicit pro-inflammatory and pro-thrombotic responses that can initiate the earliest stages of vascular disease and accelerate rates of CVD and renal disorders.

Blood levels of SAA are related to CVD risk, however, high-density lipoprotein (HDL) that binds SAA shows an inverse relationship with CVD [11]. Thus, the interaction between HDL and SAA (and indeed the HDL:SAA ratio) may be important in defining the biological actions of SAA [12]. For example, we previously determined that HDL attenuates the pro-inflammatory/thrombotic activities of SAA and, conversely, that SAA adversely affects the anti-atherogenic actions of HDL in various ex vivo endothelial cell culture models [12,13,16]. Binding of SAA to HDL can displace other apolipoproteins, particularly ApoA-I, thus diminishing HDL’s participation in anti-atherogenic lipid metabolism and transport pathways and, consequently, promoting increased endothelial proteoglycan expression, an early event in the pathogenesis of atherosclerosis [17,18]. SAA enrichment can impair anti-inflammatory properties of HDL as shown in patients with diabetic nephropathy [19] and may interfere with HDL’s modulation of pro-atherogenic modifications to low-density lipoprotein (LDL), endothelial cell adhesion molecules i.e., intracellular adhesion molecule/vascular cell adhesion molecule (ICAM/VCAM) expression and monocyte infiltration into the artery wall [20,21]. HDL also inhibits SAA-mediated reactive oxygen species generation and Nod-like receptor protein 3 (NLRP3) inflammasome activation [22]. Nonetheless, others have reported contradictory results showing that reconstituted HDL containing SAA showed higher antioxidant potential than normal HDL [23] and that SAA bound by HDL exhibited no discernible cytotoxicity under pathophysiological conditions [24].

As a consequence of the reported anti-inflammatory/antioxidant properties of HDL [20,21,25,26,27], its levels were considered to directly reflect CVD risk (HDL–cholesterol hypothesis), and hence the notion that HDL is protective against atherosclerosis gained favorable attention supported by a series of animal studies [28]. Later, this notion was challenged by human gene studies, particularly Mendelian randomization studies using mutations in ApoA-I, ABCA1 (ATP binding cassette A1), and LCAT (lecithin cholesterol acyl transferase) that manifest as low HDL concentration in the absence of significant CVD risk [29,30,31,32]. Thus, a role for HDL in CVD and as a therapeutic target in the classic HDL–cholesterol hypothesis may be refined to an HDL function hypothesis, which is currently under clinical investigation, although initial clinical trial outcomes have been disappointing [33]. Further validating studies are warranted. Thus, subject-specific HDL function that represents HDL activity in an individual, which is difficult to estimate in the clinical setting and unlikely to be reflected by a simple measurement of plasma HDL concentration, may be more pertinent to CVD risk than just evaluation of HDL–cholesterol concentration. Furthermore, use of HDL as a therapeutic agent may depend upon how alterations in circulating HDL concentration are induced (e.g., through supplementing reconstituted or native HDL). In this context the current study will test the hypothesis that administration of native human HDL will inhibit recombinant SAA’s propensity to elicit pro-inflammatory activity, stimulate oxidative damage, and enhance aortic and renal pathology in ApoE^−^/^−^ mice mouse model of atherosclerosis.

## 2. Results

### 2.1. Evidence that HDL Inhibits the Pro-Atherogenic Activity of SAA

#### 2.1.1. HDL Pretreatment Inhibits SAA-Induced Aortic Lesions

Histological analysis of sectioned aortic root 18 weeks after commencement of SAA treatment showed marked increase in atherosclerotic lesion at the aortic root compared to vehicle and LPS control groups (refer to the yellow arrows in Figure 1A–C and the corresponding highlighted areas in the representative images). Determinations of the fractional lesion area relative to the lumen of the aortic root revealed a significant increase in the percentage of lesion size in ApoE^−^/^−^ mice treated with SAA alone, which was ~6% higher than mean lesion values determined for control and LPS groups (compare Figure 1A,B and refer to Figure 1E). Furthermore, close inspection of the aortic lesion in mice stimulated with SAA showed the presence of complex late-stage lesions typified by the formation of a necrotic core containing cholesterol clefts. By contrast, mice pretreated with human HDL prior to SAA stimulation showed significantly lower percentage of lesion (HDL + SAA group) than mice supplemented with SAA alone (compare Figure 1C with Figure 1D and refer to Figure 1E) with a general absence of a highly developed necrotic core. Notably, mice stimulated with LPS showed comparable mean lesion size to the vehicle control, which suggests that the selected dose of LPS (10 pg LPS/μg SAA administered) was unable to initiate an inflammatory response that stimulated lesion development, thereby confirming the pro-atherogenic activity of recombinant SAA. As anticipated, aortic lesions in the corresponding young cohort (i.e., specimens harvested 4 weeks after commencement of treatment) were absent in regions near the base of the valve leaflets and were not different among the various treatment groups.

#### 2.1.2. HDL Pretreatment Mitigates SAA-Induced Pro-Atherogenic Changes to the Vasculature

Vascular cellular adhesion molecule-1 (VCAM-1) promotes monocyte adhesion to the endothelium in response to inflammatory cytokines. VCAM-1 is expressed early during aortic lesion development compared to intercellular adhesion molecule-1 (ICAM-1), which extends to uninvolved aorta. Therefore, VCAM-1 is considered an ideal marker to study atherogenesis [34]. Sections of thoracic aorta from control, LPS, SAA, and HDL + SAA groups (taken 4 weeks after commencement of treatment) were assessed for VCAM-1 immuno-reactivity (Figure 2panels A–D and corresponding high-magnification images panels E–H). As we described previously [35], discernibly higher VCAM-1^+^ (dark granular accumulation at the cell surface) immunostaining was observed in thoracic aortae from the SAA group (Figure 2panels C,G) than corresponding aortic tissue from the vehicle control (Figure 2panels A,E) and LPS groups (Figure 2panels B,F) that showed virtually no VCAM-1^+^ staining of the endothelium. By contrast, mice treated with HDL before stimulation with recombinant SAA (HDL + SAA group) showed weak VCAM-1^+^ immunostaining comparable to the vehicle and LPS controls (Figure 2D,H), suggesting that HDL inhibited the expression of the adhesion protein on the vascular endothelium.

Next, we analyzed the gene expression levels of VCAM-1 in aortic homogenates for the corresponding aorta harvested 4 weeks after commencement of treatment, a period of time suitable to assess early vascular dysfunction. Consistent with increased immune reactivity for VCAM-1, the gene expression of VCAM-1 in the SAA-treated mice increased 50% (reaching ~2.1-fold higher than in the corresponding vehicle control), although this did not reach statistical significance. In mice treated with human HDL prior to stimulation with SAA, the extent of VCAM-1 gene expression was similar to control and LPS groups, although once again this did not reach statistical significance compared to mice treated with SAA alone (Figure 3A).

Enhanced vascular VCAM-1^+^ immune-reactivity in response to SAA is indicative of endothelial dysfunction and this may manifest as decreased bioavailability/bioactivity of vasodilating nitric oxide (NO). Therefore, we next measured cyclic guanosine monophosphate (cGMP) as a surrogate marker for the endothelial NO-soluble guanylyl cyclase-cGMP vasomotor axis. Overall, cGMP levels were significantly diminished in the aortae homogenates collected from the young cohort following treatment with SAA alone compared to control and LPS groups, as we previously reported [35]. Notably, HDL pretreatment prevented SAA-stimulated loss of cGMP with aortic cGMP levels increasing to similar levels detected in the vehicle and LPS controls (Figure 3B).

### 2.2. HDL Pretreatment Inhibits Oxidative Lipid Damage in the Vasculature

Continuing with the notion that HDL possesses anti-inflammatory and antioxidant properties we measured F_2_-Isoprostane levels in the aortic homogenates from the mice sacrificed at 4 (young cohort) and 18 weeks (old cohort) after commencement of treatment as a marker of vascular oxidative stress. Levels of aortic total F_2_-Isoprostane were significantly higher in mice treated with SAA alone when compared to corresponding values determined in aortae from corresponding vehicle and LPS controls (Figure 4A,B). This remained true irrespective of whether tissue was taken from the young or old cohorts, as we reported previously [35]. Notably, pretreatment with human HDL significantly inhibited the formation of aortic F_2_-Isoprostane in both groups of mice (Figure 4A,B), indicating that lipid oxidation was diminished in the presence of the supplemented lipoprotein. Of note, the SAA-mediated increase in oxidative lipid was more pronounced in younger mice compared to aortae isolated from older mice, suggesting that the acute inflammatory response leading to leukocyte accumulation and activation in the vasculature may be responsible for the increased oxidative damage in the early stages poststimulation with recombinant SAA.

### 2.3. Analyses of Heart Tissues

#### 2.3.1. HDL Pretreatment Mitigates SAA-Induced Pro-Atherogenic Cardiac Vasculature

The extent of endothelial activation in the coronary circulation was assessed by VCAM-1 expression in heart tissue. Consistent with the data generated from aortic vessels, VCAM-1 mRNA expression in heart tissue increased significantly in the SAA treatment group compared to mice treated with vehicle or LPS as controls (Figure 5). Consistent with HDL inhibiting SAA bioactivity, VCAM-1 mRNA expression decreased in the HDL + SAA group when compared to SAA alone in the young mouse cohort (Figure 5).

#### 2.3.2. Inflammatory Cytokines Expression

In addition to VCAM-1, other pro-inflammatory markers were assessed in myocardial homogenates. Overall, TF, TNFα and NFκB mRNA expression was significantly increased in mice treated with SAA alone compared to vehicle and LPS control groups, as previously described [35], while once again pre-supplementation of mice with HDL (HDL + SAA) markedly decreased the expression of these selected inflammatory cytokines/chemokines and prothrombotic tissue factor to near control levels determined in the young cohort (Figure 6A–C).

### 2.4. Kidney Function Studies

#### 2.4.1. SAA Stimulates Renal Vascular Endothelium Dysfunction and HDL Mitigates These Changes

The kidneys are a highly vascularized organ and prone to altered blood flow typically elicited by changes in vascular tone. As SAA promotes vascular dysfunction and altered vascular NO bio-activity (as demonstrated above), it follows that organs such as the kidney may be impacted by SAA stimulation. To assess renal function, urine samples were collected from mice culled after 4 (young group) and 18 weeks (old group) after the commencement of the various interventions and subsequently analyzed for total protein, as we reported previously [35]. In the young cohort (Figure 7A), significantly higher levels of albumin were found in the urine of mice assigned to the SAA group compared to vehicle-control, whereas urinary protein only trended to be higher in all groups from the old cohort compared to respective control, and this difference between SAA and controls in the older mice was not statistically significant (Figure 7B). Notably, added human HDL significantly inhibited the SAA-stimulated increase in urinary protein in samples from the young cohort (Figure 7A), indicating HDL preserved renal function in these mice.

#### 2.4.2. HDL Pretreatment Protects Renal Tissues from SAA-Induced Acute Injury

The biomarker KIM-1 is useful in evaluating acute kidney injury in response to renal insult. Overall, urinary KIM-1 in the young cohort was significantly higher in SAA-treated mice than that determined in urine samples from the control and LPS groups, as we described previously [35]. Consistent with other biomarkers of renal damage measured here, HDL treatment effectively preserved the renal tissue from SAA-induced damage, as indicated by a marked diminution of KIM-1 in mice pretreated with human HDL before administration of recombinant SAA (Figure 8A). In addition to measurement of this biochemical measure of renal tubule damage, we also assessed changes to renal glomeruli in the different treatment groups (Figure 8B). The data indicate that administered LPS marginally decreased Bowman’s space (decreased 10% vs. vehicle control, *p* < 0.05), whereas Bowman’s space was more markedly decreased (decreased 15% vs. vehicle control, *p* < 0.05) in the presence of SAA. Notably, HDL coadministration prevented this SAA-stimulated decrease in Bowman’s space (decreased 3% vs. vehicle control, *p* < 0.05) with glomeruli appearing less condensed and more similar to corresponding glomeruli in the control.

#### 2.4.3. Human HDL Modulates SAA-Mediated Oxidative Stress in Renal Tissues

Upon activation, immune cells can produce myeloperoxidase (MPO)-derived oxidants such as hypochlorous acid (HOCl) [36]. This potent two-electron oxidant can induce uncontrolled oxidative damage to tissues. Renal oxidative damage was assessed in renal tissues using 3-Cl-Tyr as a specific biomarker for HOCl-mediated protein oxidation [36]. As we described previously [35], significantly higher levels of 3-Cl-Tyr were observed in young as well as old mice assigned to the respective SAA treatment groups when compared to respective age-matched controls (Figure 9). Interestingly, the ratio of 3-Cl-Tyr/native Tyr was more markedly elevated in the acute response to SAA at 4 weeks than at 18 weeks (A “young mice” compared to B “old mice”). In the presence of supplemented HDL, the SAA-stimulated increase in 3-Cl-Tyr/native Tyr was significantly inhibited, more so in the young than the older mice, once again indicating that added HDL suppressed SAA pro-inflammatory activity.

#### 2.4.4. Pretreatment with HDL Inhibits SAA Induced Pro-Inflammatory Cytokine Stimulation

Next, we investigated selected responses for cytokines (IL- 1α, IL-1β, IL-2, IFN-γ, MCP-1 and GMCSF) in renal tissue from the young cohort using a multiplex ELISA approach where the selected inflammatory markers were simultaneously assessed in the same tissue homogenate, as we described previously [35]. Comparing kidneys from the young cohort it, was demonstrated that all selected cytokines/chemokines increased significantly after treatment with SAA alone, relative to the vehicle and LPS controls (Figure 10). Notably, this SAA-induced over-expression of selected cytokines was generally prevented with HDL pretreatment, suggesting that HDL can substantially ameliorate SAA-induced pro-inflammatory responses in the kidney, a result completely consistent with the demonstrable decrease in the protein oxidative damage marker 3-Cl-Tyr/native Tyr in the same kidney tissue (Figure 9, above).

#### 2.4.5. SAA Stimulates p-38/MAPK Activation

Elevated levels of pro-inflammatory cytokines and increased oxidative damage can promote activation of intracellular kinases [37]. To evaluate renal kinase activity, the level of mitogen-activated protein kinase p38 phosphorylation (p-p38MAPK) was determined using an immune-fluorescent approach (Figure 11). Overall, all tissues showed some level of P38MAPK activation, largely localized to the intracellular domain of epithelial cells within proximal tubules in the renal cortex with little evidence of p-p38MAPK^+^ immune-positive response within glomeruli (glomerular structures are indicated by orange arrows in all panels as defined by the cluster of DAPI positive cells). Notably, no p-p38MAPK^+^ immune-positive response was evident in the corresponding negative control (refer to the insets shown in Figure 11), indicating that the antibody targeted the correct antigen. In mice treated with SAA, the level of renal p-p38MAPK^+^ immuno-reactivity increased relative to the vehicle and LPS controls, and this was more marked in the renal tissue from older mice compared to that from young mice. Notably, animals pretreated with HDL prior to SAA stimulation (HDL + SAA group) showed variable impact on p-p38MAPK^+^ immuno-reactivity. Specifically, in the older mice the presence of elevated HDL diminished fluorescence intensity to below the level detected in the corresponding age-matched vehicle and LPS controls, whereas in the younger mice the inhibition was less marked (Figure 11).

## 3. Discussion

Chronic, low-grade inflammatory conditions such as diabetes, rheumatoid arthritis, obesity, and atherosclerosis are characterized by elevated circulating levels of SAA that is associated with increased risk of cardiovascular disease (CVD) [38]. Previously, we demonstrated that SAA administration leads to vascular inflammation, enhanced formation of atherosclerotic lesion, and renal dysfunction in ApoE-deficient mice. The ApoE-deficient mouse is an established murine model for human atherosclerosis [35]. It follows that SAA may play a role in accelerating CVD in ApoE-deficient mice, which is dependent on the ratio of HDL/SAA in circulation. Herein, we documented that co-supplementing ApoE^−^/^−^ mice with SAA and HDL inhibited SAA-mediated pro-inflammatory and prothrombotic responses in vascular, cardiac, and renal tissues, and prevents SAA-stimulated enhancement of aortic lesion. Consistent with a pro-atherogenic activity, SAA stimulated VCAM-1 gene and protein expression in aortic vessels, and mice administered SAA showed increased lesion size at the aortic root and levels of aortic F_2_-isoporstane as a marker of oxidative damage. In heart tissue, SAA induced significant upregulation in mRNA for VCAM-1, the transcription factor NFκB, the inflammatory cytokine (TNFα), and procoagulative tissue factor (TF). SAA also caused enhanced inflammation in the kidneys as reflected by significant increases in selected cytokine protein accumulation (IL- 1α, IL-1β, IL-2, IFN-γ, MCP-1, and GMCSF), and a decrease in Bowman’s space and p-38MAPK activation, which manifested as increased renal injury as judged by a significant increase in urinary levels of albumin and KIM-1 and renal oxidative damage (3-Cl-Tyr). By contrast, coadministration of HDL with SAA significantly mitigated SAA-stimulated vascular, cardiac, and renal pathologies, indicating that the lipoprotein plays a significant role in altering SAA bioactivity. Therefore, the relative levels of circulating HDL and SAA may be important in determining risk of developing cardiovascular and renal disorders and interventional measures to increase HDL may limit the pro-inflammatory bioactivity of SAA in humans and ultimately decrease risk of developing these (vascular) pathologies.

Because a high-fat diet can elicit significant vascular inflammation and potentially mask subtle changes induced by SAA administration, we studied animals that were fed a standard chow so that pathologies determined herein can be attributed to SAA bioactivity rather than excessive fat loading as indicated previously [35]. Increases in VCAM-1 protein expression as early as 4 weeks (2 weeks post-SAA treatment) and subsequent increased aortic lesion size (at 18 weeks after commencement of SAA treatment) suggest that an acute increase in SAA triggers enhances pro-atherogenic factors that negatively impact the vascular endothelium and translates to accelerated atherosclerosis in these mice in the absence of high-fat loading. These data are consistent with a previous independent study [7] showing that a brief in situ increase in SAA stimulates pro-atherogenic changes to the vessel wall in mice. Furthermore, a direct role for SAA in stimulating atherogenesis in ApoE^−^/^−^ (12–16 weeks post-SAA treatment) [7,39] and LDL receptor-deficient mice [40] is already established. Thus, elevated SAA is capable of promoting atherosclerotic lesion development in mice and this is likely a result of SAA-mediated inflammation to the vascular endothelium. Whether HDL that is known to bind SAA can prevent this SAA-stimulated acceleration of atherosclerosis in ApoE^−^/^−^ mice was the focus of this study.

Regulatory effects of HDL supplementation on SAA’s pro-inflammatory bioactivity, including mitigation of production of reactive oxygen species, has been reported in isolated murine macrophage cultures [22,41]. However, in vivo studies on the role of HDL/SAA interactions in complex biological environments are lacking. Herein, we demonstrated for the first time that supplementing human HDL directly into the vasculature strongly inhibits SAA-stimulated pro-atherogenic changes in the aortae of ApoE^−^/^−^ mice fed a normal diet. Thus, our data demonstrate that pretreatment of human HDL prior to administration of recombinant SAA offered protection against SAA-induced aortic lesion development through a mechanism involving decreased VCAM-1^+^ immunoreactivity on the vascular endothelium. This diminished level of adhesion molecule expression was accompanied by improved levels of aortic cGMP, indicating that supplemented HDL rebalanced NO bioactivity that was suboptimal in the presence of SAA alone. This rebalancing likely involves a decrease in factors that consume bioactive NO, although we cannot discount that NO production was increased in the presence of the added lipoprotein. Recent studies have questioned the notion that HDL is protective against CVD, a conclusion based on studies where HDL deficiency does not appear to increase CVD risk [29,30,31,32]. This raises the possibility that not only lipoprotein concentration (HDL or SAA) is crucial but the HDL:SAA ratio and, importantly, interactions between HDL/SAA are critical determinants that can influence vascular disease progression.

F2-Isoprostane lipid peroxidation products serve as reliable biomarkers of free radical-mediated oxidative damage in vivo [42,43] and show a strong association with the development of atherosclerotic lesions [42]. Consistent with the notion that HDL possesses antioxidant properties, a significant reduction in F_2_-Isoprostane concentration in aortic homogenates from mice pretreated with HDL indicates that HDL mitigates vascular oxidative damage; HDL is the major lipoprotein transporter of plasma F_2_-Isoprostanes. Thus, under conditions employed in this study, SAA-mediated increases in oxidized lipids was prevented by HDL treatment as indicated by the lower levels of aortic F_2_-Isoprostanes in mice cosupplemented SAA and HDL and this was associated with decreased vascular lesion. To the contrary, De Beer et al. reported that endogenous SAA concentration has no impact on the development of mature atherosclerotic lesions in ApoE^−^/^−^ SAA double knockout mice fed a normal diet [44]. However, this does not negate the role for SAA to enhance atherosclerotic lesion development, as is the case determined here and reported previously by others [7,39].

In addition to SAA-induced vascular pathology, we investigated whether an acute increase in SAA leads to systemic inflammatory responses involving heart and kidneys. Consistent with the demonstrable increase in aortic inflammation, expression of VCAM-1 gene, TNFα, NFκB, and TF mRNA was higher in cardiac homogenates from mice treated with SAA, suggesting that, in addition to vascular smooth muscle and endothelial cells, cardiac myocytes might also be responsive to SAA-stimulated pro-inflammatory activity. However, this observation is not corroborated by available evidence in the literature and interpreting this data is complicated by the presence of vessels, connective tissue, and possibly blood cells (albeit, the heart was perfused prior to harvest) in the isolated heart tissue that may respond to SAA pro-inflammatory activity and, therefore, further investigation is required to unambiguously prove that cardiomyocytes are responsive to SAA.

Previously, we reported that short-term treatment of ApoE^−^/^−^ mice with SAA causes renal impairment even when mice were fed a normal diet as judged by the significant elevation of several pro-inflammatory cytokines (IL- 1α, IL-1β, IL-2, IFN-γ, MCP-1, and GMCSF) in renal tissue from mice treated with SAA alone [35]. Inflammation is one of the contributing factors in developing diabetes as demonstrated by higher levels of SAA and several other inflammatory markers such as TNF, C-reactive protein (CRP), and MCP-1 that parallel increased urinary albumin secretion in such subjects [7]. Presently, KIM-1 is considered a reliable marker of kidney damage and micro-albuminuria is an established risk factor for renal dysfunction that is linked to cardiovascular pathology [45,46]. However, results obtained here extend our previous work to now show that pretreatment with HDL followed by SAA effectively preserves renal function as indicated by decreased levels of urinary albumin and KIM-1 in mice co-supplemented with HDL and SAA. Interestingly, abnormal serum SAA levels are predictive of end-stage renal disease (ESRD) and higher mortality in diabetic kidney disease patients [14] but not in patients with type 2 diabetes without manifest kidney disease [47]. To our knowledge, this is the first study investigating the protective role of HDL against SAA-induced kidney damage.

Circulating SAA also correlates with the progression of diabetes in patients; a disorder where subjects also suffer kidney disease as a comorbidity [48]. This correlative link between diabetes, kidney disease, and elevated circulating SAA suggests that SAA may play a role in stimulating both primary and secondary pathologies in these patients. Whether the intrinsic anti-inflammatory action of HDL is responsible for modulating cytokine levels in renal tissues or whether the capacity for HDL to bind SAA and down-regulate SAA-elicited pro-inflammatory bioactivity in the kidney remains unclear.

Increased oxidative stress and elevated levels of pro-inflammatory cytokines can activate intracellular kinases [37]. Consistent with this notion, P38 mitogen-activated protein kinase (p38MAPK^+^) immunofluorescence was increased in SAA-treated mice compared to vehicle and LPS controls while pretreatment with HDL attenuated both oxidative damage (3Cl-Tyr) and p38MAPK^+^ immunoreactivity in renal tissues. P38MAPK is central to an intracellular signal transduction pathway involved in the activation of profibrotic and pro-inflammatory mediators [49]. Activation of p38MAPK induces fibrosis in cardiac [50], pulmonary [51,52], and peritoneal membranes [53]. Further, p38MAPK^+^ immune-reactivity is shown to induce renal interstitial fibrosis in mice leading to proteinuria and renal tissue inflammation [49], similar to the observations determined herein. Interestingly, pharmacological inhibitors of p38 MAPK (SB-731445, FR167653) effectively reduce its phosphorylation and ability to induce fibrosis in renal tissues [49,54], implicating this pathway in renal tissue fibrosis. In our experiments, this was evidenced by marked increases in p38MAPK phosphorylation in renal epithelial and interstitial cells from mice treated with SAA and assigned to the old cohort, which aligns with the time where enhanced fibrosis is anticipated. However, p38MAPK activation was also significantly elevated in the acute phase post-SAA treatment (young cohort). Once again, HDL treatment diminished p-p38MAPK^+^ immune-reactivity to a level lower than the respective controls in old mice and, while a discernible reduction was noted in younger mice, this was less consistent. Nonetheless, the data show that SAA is able induce activation of p38MAPK pathway in renal tissue, which may lead to fibrosis and HDL-treatment inhibited activation of intracellular stress markers normalizing proteinuria and renal tissue inflammation as demonstrated by the improvement in all pro-inflammatory and damage markers.

Concomitant with increased pro-inflammatory cytokines, significantly higher levels of 3-Cl-Tyr were detected in renal tissue from mice treated with SAA, albeit that 3-Cl-Tyr levels were higher in mice from young compared to older cohorts. One potential explanation for this difference could be the pathophysiological stage of inflammation. The acute model represents severe inflammation after SAA treatment, which stimulates a more intense immune response leading to high MPO activity and consequently elevated tissue damage. Such inflammation resolves and possibly dissipates over time (as judged at 18 weeks after commencement of SAA stimulation), reducing the immune response and subsequently the MPO activity. Hence the comparative amount of 3-Cl-Tyr likely reflects the stage and level of inflammation post-SAA treatment. Concomitant with aortic and cardiac markers, HDL treatment significantly reduced the levels of markers of oxidative damage and pro-inflammatory cytokines in renal tissue homogenates.

Under inflammatory conditions, SAA displaces ApoA-I, the predominant apolipoprotein of HDL. Although debated in the literature [23,55], at least some of the available data [56,57,58] indicate that this mode of ApoA-I displacement alters HDL function and effectively promotes a pro-atherogenic phenotype at a threshold when SAA constitutes more than half of the HDL protein [59]. Therefore, it is hypothesized that the ratio of SAA/HDL is important for HDL anti-inflammatory and antioxidant activities in biological systems. Thus, not only circulating HDL levels but the SAA:HDL ratio may be important to clinical outcome, and this may provide an explanation as to why patients with normal HDL levels remain at risk of developing CVD. Given the strong SAA binding affinity to HDL, we speculate that HDL sequesters (or mops up) free SAA from the circulation, thus limiting SAA bioactivity, although we cannot discount that the anti-inflammatory action of HDL alone (in the mice supplemented with HDL) may account for the protective action or, indeed, a combination of binding SAA and anti-inflammatory action that explains the protective effect of cosupplementing SAA and HDL. In addition, given that SAA is an apoprotein able to self-assemble with lipid, it has been purported to provide a mechanism to remove potentially bioactive lipid oxidation products [60]. Whether this is relevant to the animal model studied here is unclear. However, the pro-atherogenic action of SAA appears to outweigh any beneficial effect from any putative formation of SAA-lipid complexes [58] in this mouse model of vascular disease.

### Study Limitations

Our results show for the first time that SAA-induced renal pathologies can be protected by HDL pre-administration. However, the exact mechanism by which HDL interacts with SAA to perform this function was not investigated in this study and presents as a limitation that warrants further research to establish the importance of the relative ratio of SAA and HDL in individuals as a biomarker for pathological changes to the vasculature. Thus, while we speculated the binding of added SAA by co-administered HDL is a mechanism that explains the protective action of HDL, we have not provided evidence that this occurs in vivo. To address this, it may be possible to label SAA and track the labelled protein and test whether it can bind to HDL isolated from the circulation. However, this remains outside the scope of the present study and, although warranted to confirm the proposed mechanism, this proposed mechanism for HDL interaction with SAA remains to be established and our proposal that SAA binds to HDL is based solely on literature evidence. Furthermore, whether administered recombinant SAA stimulates endogenous SAA production and the half-life of the supplemented recombinant SAA in the mouse circulation has not been explored and this is an additional limitation of the current study.

We note that while it is common practice to use genetically modified mice as an experimental model of atherosclerosis, marked variations in outcomes between laboratories and a general lack of clinical translation of outcomes from mouse models to humans can complicate the available literature and future studies derived from this literature base. We acknowledge the statement of practice as set out in the “Recommendation on Design, Execution, and Reporting of Animal Atherosclerosis Studies: A Scientific Statement from the American Heart Association” [61] and identify the following in relation to the study reported here:We used a common genetically modified mouse model to assess atherosclerosis and accept that the shortcomings of this model, as identified in the statement, are a general limitation of this experimental model.Further validating experiments using nonrodent-based animal models (such as rabbits, pigs, and nonhuman primates) are absolutely required before the conclusions drawn from this study can be translated to human conditions where elevated SAA levels may impact on vascular and renal function.In the experimental design, we refrained from using a high-fat diet to accelerate atherosclerosis as the hypothesis being tested was that SAA itself plays a role in promoting pro-atherogenic factors that accelerate atherosclerosis and so a high-fat diet would interfere with this assessment.We used a reputable supplier of ApoE^-/-^ mice in Australia (Animal Resources Centre (Perth, Western Australia)) that supply a range of mice for research purposes, and mice were contained in the same environment within the animal facility with the access to the same chow and water supply.Mice (8 weeks old) were transported to a local site for husbandry, allowed to acclimate, and then were randomly divided into four groups without internal bias.Data obtained using (1) analytical and (2) imaging techniques reported in this study were repeated using the same tissues at the same time for all of the treatment cohorts to gain a valid and rigorous comparison between vehicle, LPS, SAA, and HDL-intervention cohorts.We reported how often a given experiment was repeated to substantiate the outcome, and this was established using the nominated statistical tests with appropriate corrections.

## 4. Experimental Section

### 4.1. Materials

Recombinant human serum amyloid A (SAA) was obtained from a single supplier (PeproTech, SF, USA) and a single-batch preparation was used for the entire study. Lipopolysaccharide (LPS), phosphate buffered saline (PBS), and GeneElute Total RNA Miniprep Kit were from Sigma (Sydney, Australia). Fresh stock solutions of SAA were prepared in sterile-filtered PBS to yield 1 mg SAA protein/mL final stock concentration, aliquoted into capped vials (200 µL), and stored at −80 °C. As we indicated previously [35], individual vials of recombinant SAA were thawed at 22 °C before use. Levels of LPS in stock solutions of recombinant SAA stock were routinely determined using limulus amebocyte lysate (LAL) and visualizing agent Spectrozyme LAL (American Diagnostica, Stamford, CA). Overall, it was determined that the concentration of minor contaminating LPS within the preparations of SAA that were routinely used here was <2 pg LPS/µg SAA protein/mL. Thus, the recombinant SAA employed in this study contained low-level LPS contamination, which has been demonstrated to be relatively inert and unable to stimulate pathophysiologically meaningful pro-inflammatory/pro-coagulant responses in ex vivo studies with isolated human peripheral blood monocytes, a cell type known to be highly responsive to LPS stimulation [62]. However, to account for the possibility that this low-level contamination in the various preparations of recombinant SAA is responsible for eliciting inflammation in vivo, a second LPS control arm was added to the design of the animal studies undertaken as we have described previously [35]. Notably, recombinant SAA was not regarded as a physiological analogue of hepatic SAA as it represents a hybrid molecule of isoforms SAA1 and SAA2. While laborious isolation and purification to remove LPS contamination of isolated human or murine SAA from blood is feasible, the difficulties in acquiring suitably pure quantities for use in the planned in vivo studies here ruled out this mode of isolation and, despite its limitations, the recombinant source was used as an experimental model for SAA protein.

### 4.2. Methods

#### 4.2.1. Isolation of Human HDL

Human HDL was isolated from freshly sourced human whole blood as required. Briefly, blood (50 mL) was taken from healthy overnight fasted subjects and collected into heparin vacutainers (Becton, Dickinson and Company, Singapore). Blood was centrifuged (AllegraTM 6R centrifuge, Beckman, USA) at 1430× *g* for 15 min at 4 °C to obtain fresh human plasma. Subsequently, HDL was isolated from plasma by a second density-gradient ultracentrifugation step, as described in detail elsewhere [63]. Specifically, plasma (volume ranging 0.4−1.9 mL) was adjusted to 1.24 g/mL density by dissolving of 381.6 mg KBr/mL plasma before ultra-centrifugation at 356,000× *g* at 15 °C for 3 h with a Beckman benchtop ultracentrifuge (Beckman Optima TLX, USA). The ultracentrifuge was equipped with a TLA 100.4 rotor to accommodate 1.9 mL plasma in 5.1 mL quick-seal plastic tubes (Beckman Coulter, Indianapolis, IN, USA) with plasma over-layered with buffer. After heat-sealing, the tubes were centrifuged to reveal the different lipoprotein fractions as separated, tight, yellow-orange colored bands above the lipoprotein-deficient plasma fraction.

The resultant HDL band was aspirated using a 21G 1^½^ needle attached in a 1 mL syringe by direct puncture of the tube and siphoning of the band that was visible above the pool of lipoprotein-deficient plasma at the bottom of the tube. Low-molecular weight, water-soluble compounds present in the HDL fraction were removed by gel filtration using a NAP−10 column (GE Healthcare, Uppsala, Sweden) that was activated with PBS (pH 7.5; 3× bed volume). This gel-filtration procedure removed any residual low-molecular weight antioxidants including vitamin C/plasma glutathione from the lipoprotein fraction [64] so that when this HDL preparation was introduced into mice there was no increase in these biological antioxidants in the circulation, which may otherwise confound interpretation of the data. All procedures were carried out on ice and with argon degassing of solutions (30 s, controlled bubbling rate) to prevent autoxidation of HDL-associated polyunsaturated lipids. Prior to use, the protein concentration of the HDL stock solution was determined using a bicinchoninic acid assay (Thermofisher Scientific, Rockford, IL, USA) and the remaining HDL was stored at 4 °C in the dark for use within 24 h of isolation.

#### 4.2.2. Animals

All mouse studies were conducted with appropriate local ethics approval (Sydney South West Area Health Services, AEC approval #2013/041, September 1, 2013). Male apolipoprotein E deficient (ApoE^−^/^−^mice (7 weeks of age) were obtained from the Animal Resources Centre (Perth, Western Australia). All animals were acclimated to the local environment at the Heart Research Institute Animal Facility for one week prior to study commencement. Mice were housed in groups of 4 mice/cage, maintained at 22 °C with 12-h light-dark cycle with water and normal chow (cat#23200-12152, Specialty Feeds, Glen Forrest, WA 6071) provided ad libitum. Studies supplementing SAA in the presence or absence of human HDL were conducted in the absence of a high-fat diet, as under these conditions SAA stimulates a subtle inflammatory response [35] that may be masked by high fat-stimulated inflammation.

#### 4.2.3. Experimental Groups

Male mice (8 weeks old) were randomly divided into four groups and the study conducted simultaneously with all mouse cohorts studied concurrently as follows:(1)Vehicle control group: Mice received 100 µL of sterile saline via intraperitoneal (i.p.) injection route every third day for 14 days.(2)LPS group: Mice received LPS (equivalent to 25 pg LPS/kg) via i.p. injection route every third day over 14 days. This second (positive) control was included at a slightly higher concentration of LPS determined in the SAA preparation to rule out whether biological effects induced by SAA could be attributed to the LPS contaminant in the SAA protein preparation.(3)SAA group: Mice received 100 µL of SAA (stock solution 120 μg SAA/mL, total SAA 10 μg/kg mouse/injection) via i.p. injection route every third day for 14 days.(4)HDL group: Mice designated to receive HDL supplements were pre-injected with 100 μL of stock purified human high-density lipoprotein (HDL) (freshly isolated human HDL preparations were diluted in sterile PBS to yield a stock concentration of 1 mg HDL protein/mL, total 100 μg HDL protein/per kg mouse) every third day via tail vein injection for 14 days prior to treatment with SAA (as described above in group 3). In summary, mice received HDL for two weeks prior to the administration of SAA for the ensuing two weeks.

Animals were sacrificed at two time points after commencement of treatment. These time points were selected based on screening the extent of SAA-stimulated inflammation detected previously in aorta and kidney [35].

Four weeks after commencement of SAA treatment (referred to as the young group representing an acute model and judged to be a suitable time for assessing early general pro-inflammatory changes in aortic and renal tissues) and eighteen weeks after commencement of SAA treatment (referred to as the old group representing a chronic model suitable for assessing marked phenotypic changes to the vasculature including the development of quantifiable aortic lesion).

At each specified time point, biological specimens including urine, blood, heart, aortic tree (to the bifurcation distil to the renal arteries), and both kidneys were collected and processed for subsequent gene, biochemical, and histological analyses. Note, the suite of organs harvested from mice assigned to the vehicle, LPS, SAA and HDL+SAA groups were recently used to study the impact of SAA stimulation on the vascular endothelium and kidneys. These data have been reported [35]. However, the role for HDL in affecting SAA-mediated inflammation determined in parallel in the same matching ApoE^−^/^−^ mouse cohort at the same time in parallel was not reported, and this is now the focus of this current study. Thus, while the impact of SAA stimulation on vascular inflammation and renal dysfunction has been reported in ApoE^−^/^−^ mice, determination of SAA-enhanced aortic lesion formation, altered inflammatory status of cardiac tissues, and the impact of HDL supplementation on these SAA-stimulated changes to aorta and kidneys are novel and not available elsewhere in the literature to the best knowledge of the authors.

#### 4.2.4. Urine and Blood

Note, for urine collection mice were neck-scuffed and simultaneously a capped tube (2 mL, Eppendorf, Sigma-Aldrich, Sydney Australia) was placed adjacent to the penis to collect expressed urine. This process of obtaining samples generally yielded between 10–50 μL of urine that was immediately snap frozen in liquid nitrogen and stored at −80 °C for subsequent analysis.

Immediately following urine collection, animals were anesthetized under 2.5% isoflurane using a fitted rodent nose cone with the rate of oxygen-carrying gas flow adjusted using an approved commercial veterinary anesthetic machine (Tabletop Stinger system, AAS, Gladesville, Sydney Australia). After checking for suitable depth of anesthesia, a midline incision was made, and a thoracotomy was performed to expose the beating heart and the vasculature. Blood was collected by direct left ventricular puncture with a syringe containing heparin (100 IU) and immediately centrifuged (680× g) to separate the plasma, which was stored (−80 °C) and assigned for biochemical analysis.

#### 4.2.5. Collection of Aorta, Heart, and Kidney Specimens

After blood collection, the vasculature was perfused with phosphate puffer saline (PBS) administered under gravity (organ perfusion pressure was adjusted to ~80 mmHg) and the organs were resected. The upper segment of the heart proximal to the aortic root, aorta, and right kidneys were assigned to histology and fixed in 4% *v*/*v* formaldehyde solution and subsequently transferred to 70% *v*/*v* ethanol/H_2_O. One half of the right kidney and remaining heart tissue (primarily the ventricles) were snap frozen in liquid nitrogen and stored at −80 °C until required to prepare kidney/myocardial tissue homogenates used in subsequent protein and gene analysis (homogenization followed the procedure described in detail previously [35]. The remaining half kidney was fixed in 4% *v*/*v* formaldehyde solution and embedded in Tissue Plus^®^ optimal cutting temperature (OCT) compound (Fischer Scientific, North Ryde, Sydney Australia). Next, aortic roots (including the apex of each heart) were dissected from the upper heart segments and embedded by the following process: (1) 2 drops of commercial OCT solution were placed into a plastic cryomold, (2) the tissue was oriented into a preferred section plane, and, finally, (3) the remaining aortic tissue was covered with extra OCT, and subsequently snap frozen in liquid nitrogen followed by long-term storage at −80 °C prior to use in immunohistochemistry studies.

Where required, the aortae close to aortic root, emerging from the heart from the apex, were sectioned at 5 or 10 microns. Whole kidney was paraffin imbedded and sectioned longitudinally and cross-sectionally. Subsequently, sections were stained with hematoxylin and eosin (H&E) (kidney, myocardium, and aortic root), with lesion size and the extent of aortic VCAM-1^+^ immuno-staining determined as described below. Stained, thin sections (5 or 10 μm thickness) were viewed and images were captured using an Olympus Photo Microscope fitted with digital camera (DP Controller, v2.2.1.227). The captured images were converted to JPEG and analyzed for lesions size and total lumen area measurements using Image J software (v1.42, NIH, USA). In the case of captured kidney sections, a total 30 glomeruli per group (corresponding to 6 glomeruli/animal) were used to measure Bowman’s space, as described in detail previously [35]. In the case of the aortic root, care was taken to ensure that the image-captured sections contained valve leaflets at the same anatomical level in the hearts from the different treatment groups, as we have published previously [65]. This approach yielded lesion area as a relative percent of the total luminal area at the same anatomical region of the aortic root as defined by using Image J to map around the internal lumen that defined the aortic root in the section being investigated.

#### 4.2.6. Gene Expression Studies

Expression of selected genes (VCAM-1, tissue factor-TF, tumor necrosis factor alpha-TNFα, and nuclear factor kappa B (NFκB)) implicated in atherosclerosis-associated inflammation was analyzed by quantitative real-time polymerase chain reaction (q-RT-PCR) using a SensiMix SYBR kit (Bioline, Sydney, NSW, Australia). Briefly, duplicate reactions containing 2× SYBR qPCR SuperMix, 2 μL template cDNA, or distilled water and the respective forward and reverse primer sets (primer sequences summarized in Table 1) were prepared, and amplification studies conducted and quantified with a Rotor Gene 6000 (Corbett, Sydney, NSW, Australia) using standard commercial qPCR software. Relative gene expression in tissue homogenates was quantified by the comparative CT method, with the expression of the target gene normalized against β-actin (ACTB) and expressed as the fold-change relative to control (arbitrarily assigned a value of 1).

#### 4.2.7. Analysis of Inflammatory Proteins and a Biomarker of Kidney Injury with ELISA

Selected inflammatory cytokines/chemokines (including IL-1α, IL-1β, IL-2, IFN-γ, and GMCSF) were measured in protein-normalized kidney tissue homogenate using a Q-Plex mouse cytokine multiplex array kit as per manufacturer’s instructions (QuanSys Biosciences, Logan, UT, USA) as detailed previously [35]. This commercial kit was selected as the raft of chemokines and cytokines included in the kit were judged to be relevant to the putative bioactivity for SAA and were collected in a single multiplex array that allowed simultaneous assessment of all the analytes measured. Where required, urinary levels of kidney injury molecule-1 (KIM-1) were determined using a singleplex ELISA kit as per manufacturer’s instructions (ADIPO Biosciences, Santa Clara, CA, USA). All proteins were quantified using the corresponding standard curve constructed using the pure standard provided in each kit.

#### 4.2.8. Assessment of Renal 3-Chloro-Tyrosine/Tyrosine Ratio

Levels of Cl-Tyrosine (3-Cl-Tyr), a marker of oxidative stress, were assayed by liquid chromatography coupled mass spectrometry as we have previously described [35]. Briefly, 3-chloro-[^13^C9,^15^N] tyrosine and [^15^N]-tyrosine (1.5 μM) were used as internal standards and added to the samples before hydrolysis, using established protocols [66,67]. Hydrolysates were purified using solid-phase extraction columns (Supelco, Sydney, Australia) activated with 100% methanol before preconditioning (2 × 2 mL) with 0.1% *v*/*v* Trifluoroacetic acid (TFA)/H_2_O. Hydrolysates were loaded onto extraction columns with 2 mL 0.1% *v*/*v* TFA/H_2_O, eluted with 80% *v*/*v* methanol/H_2_O, dried under vacuum at 60 °C, and re-dissolved in 100 μL 0.1% *v*/*v* formic acid. Unmodified native tyrosine and 3-Cl-Tyr were detected using an LC/MS/MS/MS system (Agilent, Sydney, Australia) and quantified using calibration curves constructed with authentic Tyr and 3-Cl-Tyr and their corresponding isotopically labelled isoforms (0–500 pmol) to yield the 3-Cl-Tyr/Tyr ratio.

#### 4.2.9. Assessment of Aortic Lipid Oxidation

Determinations of aortic F_2_-isoprostanes content were conducted using stored aortic homogenates (−80 °C) and a commercially available ELISA kit (Isoprostane ELISA Kit, Cayman Chemical via the distributor Sapphire Biosciences, Redfern, Sydney, Australia), according to the manufacturer’s instructions. Under these conditions, preformed tissue F_2_-isoprostane can exist in both the free and esterified forms of this polyunsaturated lipid. To release the esterified F_2_-isoprostane, homogenized tissue samples were thawed then hydrolyzed using a strong base (10 M NaOH). Next, the reaction mixture was neutralized (with 1 M HCl) then centrifuged (3060× *g*) to obtain supernatants that included both free and esterified (or total tissue) F_2_-isoprostane. Thus, this mode of processing sample homogenate released maximal free F_2_-isoprostane for subsequent analyses. The 96-well plate was set up using the commercial instructions and the standard samples in the commercial kit were used to generate a suitable standard curve. The microplate reader was set between 405–420 nm to obtain the optical absorbance for quantitation of F_2_-isoprostanes.

#### 4.2.10. Cyclic Guanosine Monophosphate (cGMP) Assessment

Vascular cGMP is produced by the activation of smooth muscle soluble guanylyl cyclase upon stimulation with bioactive/vasodilating nitroxides (NO). Thus, levels of cGMP are a surrogate measure of NO production by the vascular endothelium. Therefore, we assessed aortic cGMP-1 levels to determine whether SAA administration interfered with NO bioavailability. Where required, frozen aortae were thawed and homogenized as described above. Tissue cGMP was then acetylated according to the manufacturer’s notes and employed as the substrate for the quantitative determination of extracellular cGMP using a commercial ELISA kit (Sapphire Biosciences, Redfern, Sydney, Australia), according to the manufacturer’s protocol. Changes in absorbance at 420 nm were recorded and cGMP quantified against the standard curve generated under identical conditions and the final expression of cGMP was further normalized to the level of protein in the corresponding homogenate with values expressed as picogram (pg) cGMP/mg protein.

#### 4.2.11. Immunohistochemistry (IHC) Studies

To explore the distribution of VCAM-1 on the endothelium, aorta samples were employed in immune-histochemical studies with appropriate isotype and negative controls to eliminate/minimize nonspecific immuno-activity. Thus, aortic thin sections were dewaxed. The deparaffinized slides were rehydrated and cycled through an antigen retrieval process by boiling in a pressure cooker in either citrate buffer (10 mM, pH 6.0) or Tris/EDTA buffer (10 mM, pH 9.0) for 10 min. To diminish endogenous peroxidase activity and the potential for nonspecific immune-reactivity, the slides were blocked by incubating in 3% *v*/*v* methanolic H_2_O_2_ and 5% *v*/*v* goat serum/PBS for 30 min, respectively. Then, slides were incubated with monoclonal anti-VCAM-1 (aorta, Santa Cruz, Sapphire Biosciences, Sydney, Australia). Alternatively, control sections were incubated with either PBS (negative control) or appropriate mouse IgG (isotype controls). All sections were then stepped to washing and incubated with biotinylated secondary anti-mouse IgG, followed by streptavidin-HRP reagent (VECTASTAINR ABC kit; Vector Laboratories, Burlingame, CA, USA) for 30 min and, finally, visualization with diaminobenzidine (DAB; DAKO Cytomation, Carpinteria, CA, USA). Sections were counterstained with Harris’s hematoxylin or imaged directly with a BX60 microscope (Olympus Australia, Notting Hill, VIC, Australia) fitted with a fluorescence source.

#### 4.2.12. Immunofluorescence (IF) Studies

The MAPKinase phosphate-P38 (p-p38) is a stress-activated kinase stimulated by oxidants such as hypochlorous acid [68]. To investigate whether P38 MAPK activation was evident in renal tissues in this study, paraffin-imbedded kidney specimens were sectioned at 5 microns and placed on frosted glass slides that were baked in the oven set to 67 °C for 1 h to allow sufficient adherence of the tissue to the slide. Heat-treated slides were then dewaxed using xylene (2× solvent changes, 10 min/change) and rehydrated in graded alcohol (100, 80, 70% *v*/*v*), incubating the slides for 2 min at each alcohol concentration. Antigen retrieval (heat retrieval) was performed in a Biocare Decloaking Chamber (DAKO Cytomation, CA, USA) for 40 min.

Next, slides were incubated in 3% *v*/*v* methanolic H_2_O_2_ and Dako protein block serum-free for 30 min in order to block endogenous peroxidase activity and nonspecific immunoreactivity, respectively. Specimens were incubated in primary antibody (dilution 1:200 *v*/*v*) for 1 h followed by envision or secondary antibody incubation for 30 min at 22 °C. After 10 min of incubation in Opal fluorophore, specimens were incubated in 0.5% *w*/*v* Sudan Black for 10 min to quench any autofluorescence followed by DAPI staining for 5 min to highlight individual nuclei. Slides were analyzed under upright fluorescent microscope (ZEISS AXIOscope, Talavera Rd, Sydney, Australia). Staining intensity was quantified by Image J software (v1.42, NIH, USA). Renal sections from control, LPS, SAA, and HDL + SAA groups were run in triplicates (*n* = 3/group). During each run, negative control slides (corresponding to treatment of the tissue with secondary antibody alone) were assessed in parallel to ensure the efficacy and reliability of antibodies and validate the immunostaining technique. Random areas from proximal renal tubules along with glomeruli were selected and imaged under upright fluorescence microscope (ZEISS AXIOscope; Talavera Rd, Sydney, Australia). Images were converted to JPEG and analyzed in Image J for quantification of staining intensity.

#### 4.2.13. Statistical Analysis

Statistical analyses were performed using the Prism statistical program (Graph Pad, San Diego, CA, USA). Data were expressed as mean ± SD and differences between data sets were determined with one-way ANOVA using Neumann–Keuls or Tukey comparison tests. Significance was accepted at *p* < 0.05.

## 5. Conclusions

Taken together our data obtained with the experimental model of atherosclerosis studied here indicate that SAA induced pro-inflammatory changes not only in vasculature and cardiac tissue but also in renal tissue and that this impaired kidney function persisted in the absence of added HDL. Furthermore, HDL which is associated with protection against CVD, has potential to mitigate pathologies that are stimulated by elevated SAA including aortic lesion formation and renal dysfunction.

## Figures and Tables

**Figure 1 ijms-21-01316-f001:**
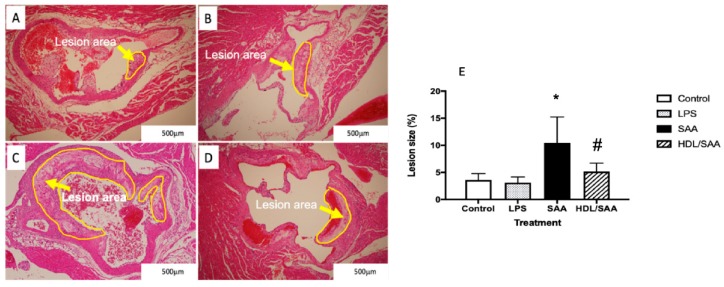
Pretreatment with HDL inhibits SAA-induced aortic lesion in ApoE^−^/^−^ mice. Male ApoE^−^/^−^ mice were randomly allocated into 4 treatment arms and administered with vehicle (control), LPS, SAA alone, or SAA in combination with HDL as described in the methods section. Mice were monitored for 18 weeks after commencement of treatment then sacrificed, and hearts and aortae were harvested. The extent of vascular lesion was expressed as the relative percentage of lesion size (determined as lesion area/corresponding luminal area at the aortic root calculated using Image J software (v1.42, NIH, USA). Representative sections show control (**A**) control, (**B**) LPS, (**C**) SAA-treated, and (**D**) isolated human HDL preloaded to animals prior to SAA treatment. Yellow arrows identify atherosclerotic lesions developing in the aortic root. (**E**) The percentage of lesion area normalized to the total area was analyzed across all treatment groups with mice numbers control (*n* = 6), LPS (*n* = 6), SAA (*n* = 6), and HDL/SAA (*n* = 6). * Different to the vehicle and LPS control groups, *p* < 0.001. ^#^ Different to the lesion measured in mice receiving SAA alone, *p* < 0.05.

**Figure 2 ijms-21-01316-f002:**
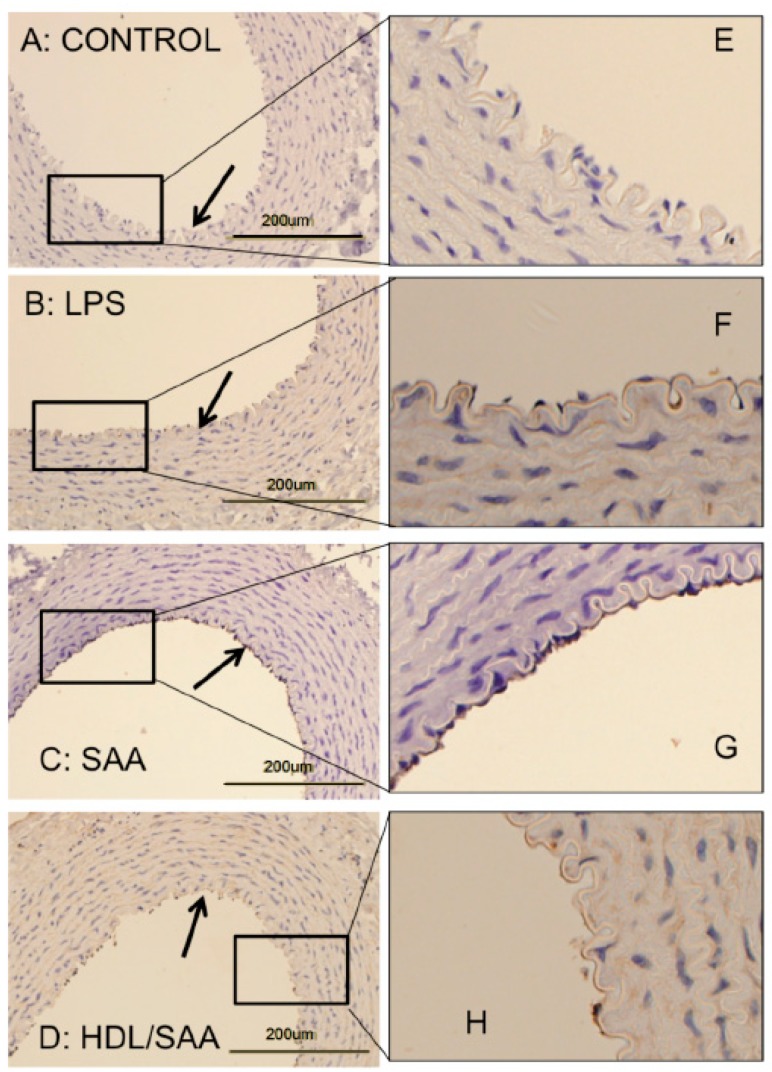
SAA enhanced atherosclerosis and induced early pro-atherogenic changes in VCAM-1 staining in the aortae of ApoE^−^/^−^ mice, which can be prevented by HDL. Male ApoE^−^/^−^ mice were randomly allocated into 4 treatment arms and administered with vehicle (control), LPS, SAA alone, or SAA in combination with HDL, as described in the methods section. Mice were sacrificed at 4 weeks, hearts perfused with PBS (50 mM, pH 7.4), then harvested. Thoracic aortae attached to the dorsal third of each heart fixed were processed for immune histological analysis. Thin sections of thoracic aorta from mice designated to the control (panel **A**), LPS (**B**), SAA (**C**) groups, as well as mice pretreated with freshly isolated human HDL for 2 weeks prior to SAA administration (**D**) were incubated with anti-VCAM-1 and imaged by light microscopy (left column). Black arrows indicate the endothelial surface on the inner lumen of the artery in panels (**A**–**D**). Dark stained immuno-reactive VCAM-1 on the vascular endothelium is evident in Panel C, whereas it is markedly lower or absent in Panels **A**, **B**, and **D**. The panels in the right column (**E**–**H**) show a high-magnification view of corresponding boxed regions of endothelial with VCAM-1^+^ immuno-reactivity seen as a granular deposit on the endothelium. Data are representative of at least 3 fields of view from each section from control (*n* = 6 mice), LPS (*n* = 6), SAA (*n* = 6), and HDL/SAA (*n* = 6).

**Figure 3 ijms-21-01316-f003:**
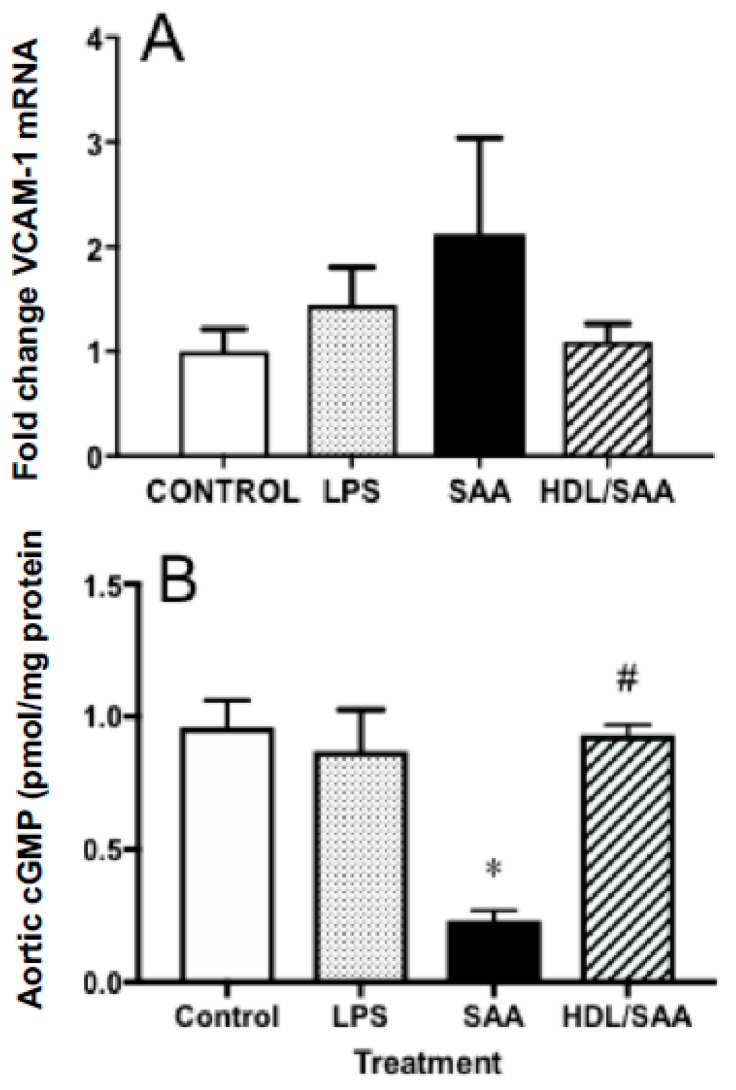
Pretreatment with HDL inhibited SAA-mediated endothelial dysfunction in ApoE^−^/^−^ mice. Male ApoE^−^/^−^ mice were randomly allocated into 4 treatment arms and administered with vehicle (control), LPS, SAA alone, or SAA in combination with HDL. Mice were monitored for 4 weeks after commencement of treatment then sacrificed, aortae were harvested, and tissue homogenates prepared as described in the methods section. The extent of mRNA expression of (**A**) VCAM-1 as a fold change and vascular levels of the secondary messenger (**B**) cGMP per protein level were determined in aortic homogenates by qPCR and ELISA, respectively. Data represent mean ± SD of control (*n* = 5), LPS (*n* = 6), SAA (*n* = 6), and HDL/SAA (*n* = 6) independent samples taken from each treatment group. * Different to the vehicle and LPS control groups, *p* < 0.001. ^#^ Different to mice receiving SAA alone, *p* < 0.001.

**Figure 4 ijms-21-01316-f004:**
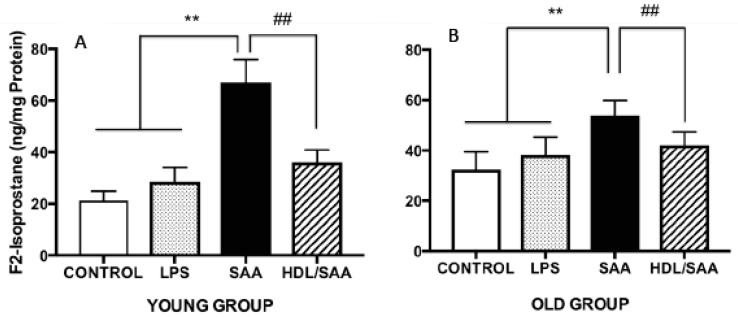
SAA-induced increase in the oxidative damage marker F2-isoporstane in aortae can be prevented by preloading with HDL. Male ApoE^−^/^−^ mice were randomly allocated into 4 treatment arms and administered with vehicle (control), LPS, SAA alone, or SAA in combination with HDL. Mice were monitored for 4 weeks (young group, panel **A**) or 18 weeks (old group, panel **B**) after commencement of treatment then sacrificed and aortae harvested. Clarified tissue homogenates were prepared and analyzed for free (unesterified) F_2_-isoprostane content per protein level by ELISA, as described in the methods sections. Each aortic homogenate obtained from control (*n* = 8), LPS (*n* = 8), SAA (*n* = 8), and HDL/SAA (*n* = 8) mice was sampled twice, and the output averaged from these 2 replicate measurements of oxidized lipid and then expressed as mean ± SD for each mouse cohort. ** Different to control and LPS group, *p* < 0.01. ^##^ Different to SAA and HDL-SAA group, *p* < 0.01.

**Figure 5 ijms-21-01316-f005:**
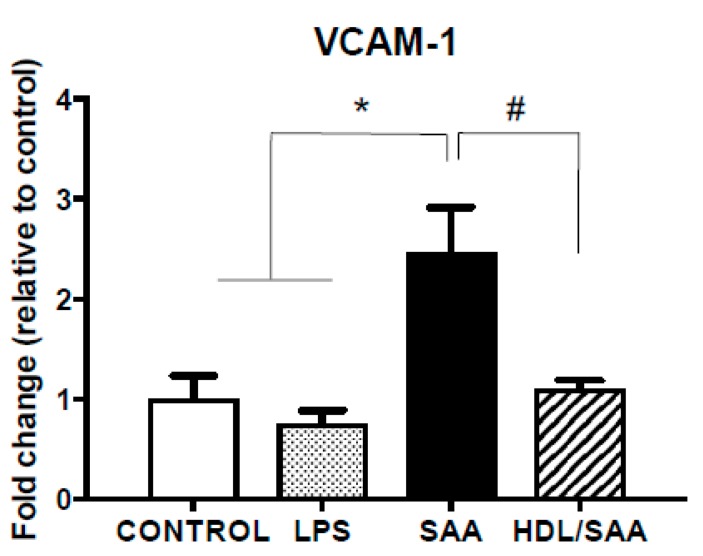
SAA induced increase in VCAM-1 expression in heart tissue is inhibited by pretreatment with HDL. Male ApoE^−^/^−^ mice were allocated into 4 treatment arms and administered with vehicle (control), LPS, SAA alone, or SAA in combination with HDL. Mice were monitored for 4 weeks after commencement of treatment then sacrificed, and hearts were harvested and processed, as described in the methods section. The extent of relative VCAM-1 mRNA expression was determined in tissue homogenates by qPCR. Data represent mean ± SD of duplicate analyses from control (*n* = 6), LPS (*n* = 6), SAA (*n* = 6), and DL/SAA (*n* = 6) mouse hearts. * Increased significantly relative to vehicle and LPS control groups, *p* < 0.001. ^#^ Decreased significantly compared to SAA alone group, *p* < 0.001.

**Figure 6 ijms-21-01316-f006:**
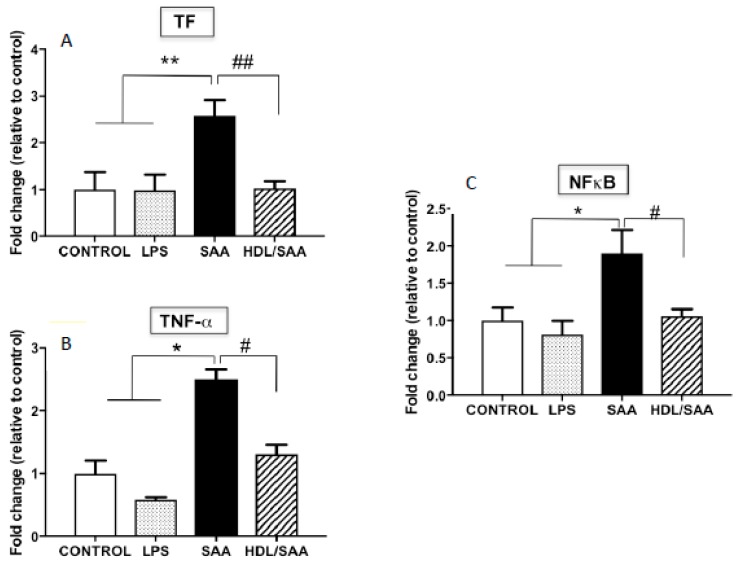
Relative gene expression of selected inflammatory cytokines (TF, TNF-α, NFκB) in mouse hearts. Male ApoE^−^/^−^ mice were allocated into 4 treatments arms and administered with vehicle (control), LPS, SAA alone, or SAA in combination with HDL. Mice were monitored for 4 weeks after commencement of treatment then sacrificed, and hearts were harvested and processed, as described in the methods section. The extent mRNA expression of TF (**A**), TNFα (**B**), and the transcription factor NFκB (**C**) were determined in heart tissue homogenates by qPCR. Data represent mean ± SD of duplicate analyses from control (*n* = 5), LPS (*n* = 5), SAA (*n* = 5), and HDL/SAA (*n* = 5) mouse hearts. * Increased significantly relative to vehicle and LPS control groups. Decreased significantly compared to SAA alone group, ** *p* < 0.001, * *p* < 0.05, ^##^
*p* < 0.001, and ^#^
*p* < 0.05.

**Figure 7 ijms-21-01316-f007:**
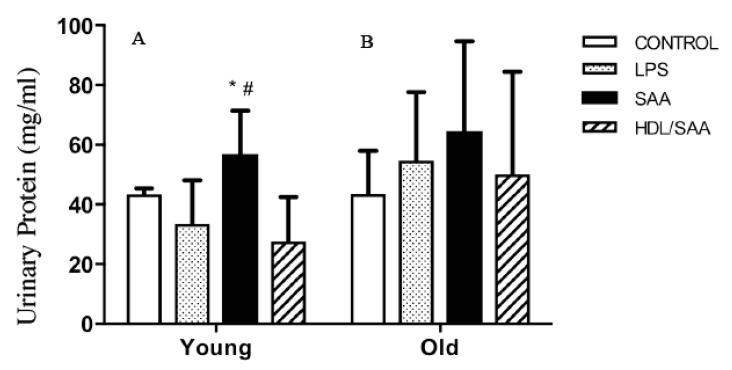
SAA induced increase in urinary protein in SAA treated mice. Male ApoE^−^/^−^ mice were allocated into 4 treatment arms and administered with vehicle (control), LPS, SAA alone, or SAA in combination with HDL. Urine was collected 4 weeks (left panel **A**) or 18 weeks (right panel **B**) after commencement of treatment and analyzed for total protein as described in methods. Data shows mean ± SD for young cohort, with mice numbers control (*n* = 7), LPS (*n* = 8), SAA (*n* = 6), and HDL/SAA (*n* = 6); and old cohort, control (*n* = 8), LPS (*n* = 8), SAA (*n* = 6), and HDL/SAA (*n* = 6). * Different to the corresponding control and LPS group, *p* < 0.05. ^#^ Different to the corresponding HDL/SAA group, *p* < 0.05.

**Figure 8 ijms-21-01316-f008:**
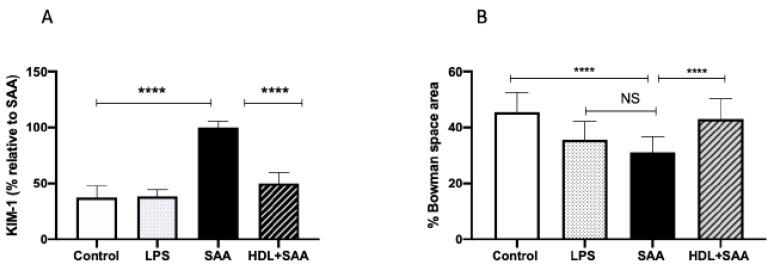
SAA-mediated increase in urinary KIM-1 was prevented by pretreatment with HDL in young mice. Male ApoE^−^/^−^ mice were allocated into 4 treatments arms and administered with vehicle (control), LPS, SAA alone, or SAA in combination with HDL. (**A**) Urine was collected 4 weeks after commencement of SAA treatment and analyzed for secretory levels of KIM-1, as described in the methods section. Data represent percentage change with respect to SAA levels where mean values were SAA = 592 pg/mL (representing 100%) with mice numbers control (*n* = 8), LPS (*n* = 8), and SAA (*n* = 8), and HDL/SAA (*n* = 8). (**B**) Thin kidney sections were stained with H&E, imaged captured, and the relative percentage area of Bowman’s space was determined using *n* = 30 glomeruli for each isolated kidney from each treatment group, as we described previously [35]. Data represent mean data from glomeruli present in the field of view (3 fields for each section) with mice numbers control (*n* = 8), LPS (*n* = 8), SAA (*n* = 8), and HDL/SAA (*n* = 8). * Different to the control and LPS group, *p* < 0.05. ^#^ Different to the SAA and HDL/SAA group, *p* < 0.05. **** *p* < 0.0001.

**Figure 9 ijms-21-01316-f009:**
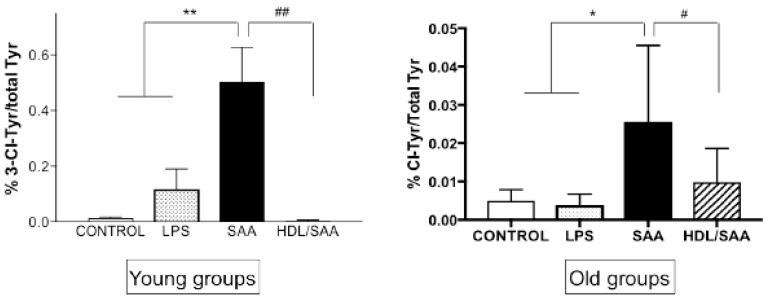
SAA-induced increase in kidney tissue 3-Cl-Tyr levels in young and old cohorts of Apo E-/- mice. Mice were allocated into 4 treatment arms and administered with vehicle (control), LPS, SAA alone, or SAA in combination with HDL. Mice were monitored for 4 weeks (young cohort) or 18 weeks (old cohort) after commencement of treatment, then sacrificed and kidneys were harvested and processed, and the levels of 3-Cl-Tyr were determined by liquid chromatography coupled with mass spectrometry, as described in methods section. Data represent mean ± SD with mice numbers were control (*n* = 5), LPS (*n* = 5), SAA (*n* = 5), and HDL/SAA (*n* = 5). Different to the respective controls, * *p* < 0.05, ** *p* < 0.01; different to the respective SAA and HDL/SAA group, # *p* < 0.05, ## *p* < 0.01.

**Figure 10 ijms-21-01316-f010:**
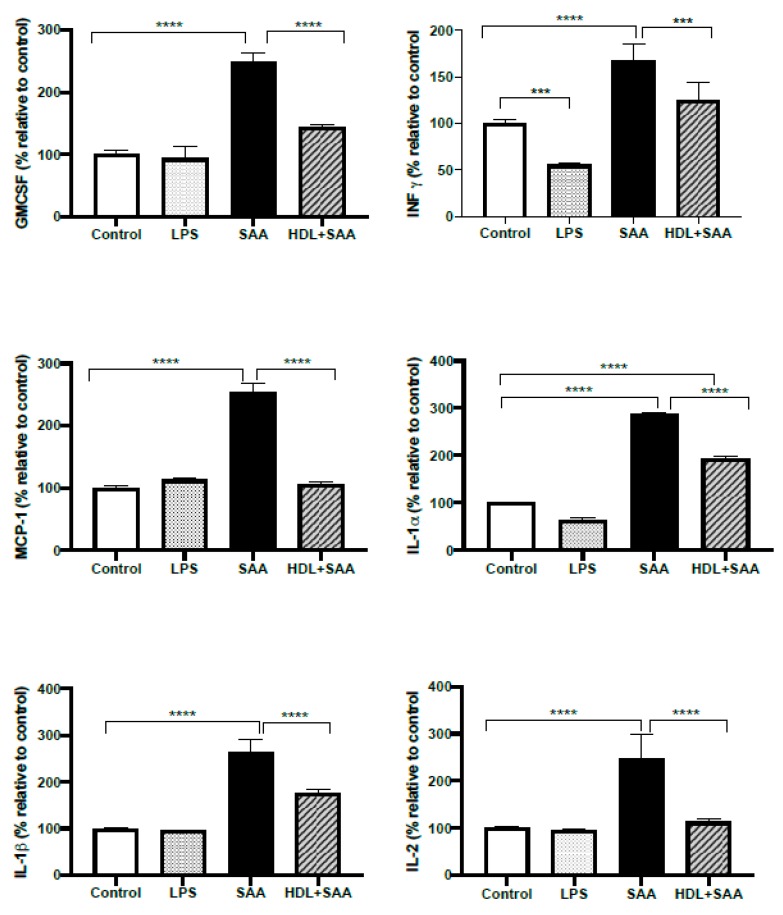
Selected cytokine levels (IL- 1α, IL-1β, IL-2, MCP-1, IFN-γ and GM-CSF) in kidney tissue from young cohort of ApoE^−^/^−^ mice following SAA and HDL treatments. Mice were allocated into 4 treatments arms and administered with vehicle (control), LPS, SAA alone, or SAA in combination with HDL. Renal tissue was harvested 4 weeks after commencement of SAA treatment, processed, and cytokines were measured by ELISA/multiplex assay in tissue homogenates, as described in the methods section. Data represent percentage change with respect to the control group (maximal SAA-stimulated values for cytokine/chemokine tested were (ng/ml): IL-1α, 21.3; IL-1β, 27.1; IL-2, 28.3; MCP-1, 25.1; INFγ, 26.1; GMCSF, 26.9). Data shows mean ± SD with mice numbers were control (*n* = 7), LPS (*n* = 8), SAA (*n* = 6), and HDL/SAA (*n* = 6). Different to the control, *** *p* < 0.001, **** *p* < 0.0001. Different to the respective SAA and HDL/SAA group, # *p* < 0.05, ## *p* < 0.01.

**Figure 11 ijms-21-01316-f011:**
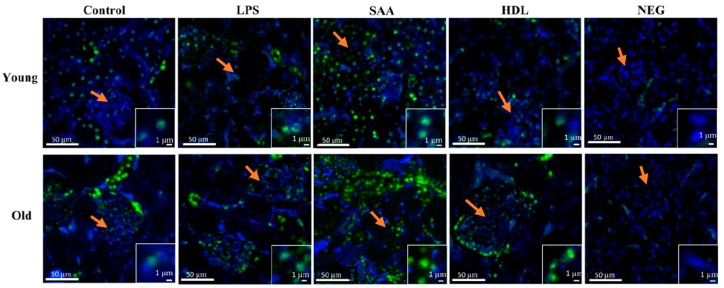
Pretreatment with HDL prevents SAA-mediated phosphorylated-p38/MAPK increases in renal tissues. Mice were allocated into 4 treatment arms and administered with vehicle (control), LPS, SAA alone, or SAA in combination with HDL. Mice were monitored for 4 weeks (young cohort) or 18 weeks (old cohort) after commencement of treatment. At each time point, mice were sacrificed, and the kidneys were harvested and processed for immunohistochemistry, as described in the methods section. After antigen retrieval, kidney sections were fluorometrically labelled with p-p38 (green) and nuclei were stained with DAPI (blue). Figures are representative of *n* = 3 fields of view taken from *n* = 3 independent samples from each treatment group (including the negative control = secondary alone) at 20× magnification. Insets to figures show higher magnification images of representative tubular epithelial cells from the different treatment groups. Orange arrows in the panels highlight glomeruli in the cortical region of each renal section.

**Table 1 ijms-21-01316-t001:** Primer pairs employed in PCR studies conducted in this study ^a^.

Gene	Forward	Reverse	NCIB Accession Number
*TNF*	5′-ATGAGCACTGAAAGCATGATCC-3′	5′-GAGGGCTGATTAGAGAGAGGTC-3′	NM_000594.4
*NFκB*	5′-CTGGAAGCACGAATGACAGA-3′	5′-TGAGGTCCATCTCCTTGGTC-3′	NM_001319226.2
*VCAM-1*	5’CCACAAGGCTACATGAGGGT-3’	5’-CAGTGTGGATGTAGCCCCTT-3’	NM_012889.1
*VEGF*	5′-TTTCTTGCGCTTTCGTTTTT-3′	5′-CCCACTGAGGAGTCCAACAT-3′	NM_001025366.3
*ACTB*	5′-CATGTACGTTGCTATCCAGG-3′	5′-CTCCTTAATGTCACGCACGAT-3′	NM_001101.5

^a^ Primer pairs were synthesized by Pro-oligo Australia and the specificity of the primers to the intended target gene was verified by a Blast search and the relevant accession numbers collated.

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
