# Peer review of "High-Density Lipoprotein (HDL) Inhibits Serum Amyloid A (SAA)-Induced Vascular and Renal Dysfunctions in Apolipoprotein E-Deficient Mice"

_ijms, 2020, doi:10.3390/ijms21041316_

Round 1

Reviewer 1 Report

The stated goal of the manuscript is to determine the effects of HDL administration on SAA-elicited inflammatory changes in mice

The stated goal of the manuscript is to determine the effects of HDL administration on SAA-elicited inflammatory changes in mice. While there are a number of strengths including the plethora of data, one of the major concerns is that the study is not clearly hypothesis driven, and the reasons for the various measurements are not clear. The authors clearly state that there is overlap between their prior publication for groups 1-3, and that the addition of group 4 is novel, but there are a number of concerns that must be clarified before this study is acceptable.

What is the hypothesis of the overall study, and how does it include both vascular and renal tissues? What is the rationale for addressing HDL impact on SAA? It is danced around, but really not clearly addressed. The rationale for the various tissues and factors measured need to be clearly addressed. At this point it seems that they just measured a whole range of things, but the link is vague at best. Peprotech SAA has been generally disregarded as it is not physiological but rather a recombinant SAA that has features of SAA1 and SAA2, and activities that were not necessarily reproduced by SAA1 or SAA2. The authors need to justify their selection of recombinant SAA for these studies as opposed to studying isolated murine or human SAA. The details re administration of HDL and SAA in group 4 is unclear. Was the SAA administered ip after each iv injection of HDL? or did mice receive 2 weeks of HDL then 2 weeks of SAA? The authors speculate that the HDL:SAA ratio may be meaningful but there is no evidence that they determined the “fate” of the injected SAA. Did it associate with HDL? Did it elevate systemic SAA in the mice (who also express endogenous SAA)? What was the half life of the injected SAA? While the LPS group did not appear to differ from the control group the injection of LPS would be expected to induce an inflammatory response in the mice including the induction of endogenous SAA. How high did the SAA go in these mice, and how long did it stay elevated? This should be compared to group 3 (SAA) and group 4 (HDL+SAA) There is no control group for HDL injection. It is possible that HDL alone could have these effects, or that HDL alone could have greater effects that were attenuated by HDL+SAA Was group 4 enrolled at the same time as groups 1-3, or were groups 1-3 a historical control? This is critical, as various studies have shown that unmeasured factors can impact outcomes – such as time of year, variations in the diet, etc There is no investigation or discussion of how HDL may attenuate SAA effects.

Minor points:

The figure legends need to be consistent – the number per group is not always stated and needs to be. Fig 1 – atherosclerosis data is usually not normally distributed and should not be presented as mean +/- SD, but generally as a dot plot. Fig 1 – the presentation of aortic root atherosclerosis as a % lesion area is atypical. What was used as the denominator (total area) and did that differ between mice – eg from variable perfusion pressure or tissue structure/artifact in section preparation. A more standard measure is to cite atherosclerotic area in um2. The authors should state if they adhered to the scientific statement on animal atherosclerosis studies by Daugherty et al (ATVB 2017). Fig 2. Is there any staining in the control group? What is the arrow indicating? Fig. 4. What does “an average of 2 replicates per sample were used” mean? How many mouse samples per group? Fig. 8, 11, and text – the term “reversed” is used inappropriately; unless the animals received SAA first and induced the outcome then received HDL the HDL can at best be stated as preventative. Fig. 10. Why is SAA stated at 100% and not the control group set as the reference group? Fig. 11. What is the NEG control?

. While there are a number of strengths including the plethora of data, one of the major concerns is that the study is not clearly hypothesis driven, and the reasons for the various measurements are not clear. The authors clearly state that there is overlap between their prior publication for groups 1-3, and that the addition of group 4 is novel, but there are a number of concerns that must be clarified before this study is acceptable.

What is the hypothesis of the overall study, and how does it include both vascular and renal tissues? What is the rationale for addressing HDL impact on SAA? It is danced around, but really not clearly addressed. The rationale for the various tissues and factors measured need to be clearly addressed. At this point it seems that they just measured a whole range of things, but the link is vague at best. Peprotech SAA has been generally disregarded as it is not physiological but rather a recombinant SAA that has features of SAA1 and SAA2, and activities that were not necessarily reproduced by SAA1 or SAA2. The authors need to justify their selection of recombinant SAA for these studies as opposed to studying isolated murine or human SAA. The details re administration of HDL and SAA in group 4 is unclear. Was the SAA administered ip after each iv injection of HDL? or did mice receive 2 weeks of HDL then 2 weeks of SAA? The authors speculate that the HDL:SAA ratio may be meaningful but there is no evidence that they determined the “fate” of the injected SAA. Did it associate with HDL? Did it elevate systemic SAA in the mice (who also express endogenous SAA)? What was the half life of the injected SAA? While the LPS group did not appear to differ from the control group the injection of LPS would be expected to induce an inflammatory response in the mice including the induction of endogenous SAA. How high did the SAA go in these mice, and how long did it stay elevated? This should be compared to group 3 (SAA) and group 4 (HDL+SAA) There is no control group for HDL injection. It is possible that HDL alone could have these effects, or that HDL alone could have greater effects that were attenuated by HDL+SAA Was group 4 enrolled at the same time as groups 1-3, or were groups 1-3 a historical control? This is critical, as various studies have shown that unmeasured factors can impact outcomes – such as time of year, variations in the diet, etc There is no investigation or discussion of how HDL may attenuate SAA effects.

Minor points:

The figure legends need to be consistent – the number per group is not always stated and needs to be. Fig 1 – atherosclerosis data is usually not normally distributed and should not be presented as mean +/- SD, but generally as a dot plot. Fig 1 – the presentation of aortic root atherosclerosis as a % lesion area is atypical. What was used as the denominator (total area) and did that differ between mice – eg from variable perfusion pressure or tissue structure/artifact in section preparation. A more standard measure is to cite atherosclerotic area in um2. The authors should state if they adhered to the scientific statement on animal atherosclerosis studies by Daugherty et al (ATVB 2017). Fig 2. Is there any staining in the control group? What is the arrow indicating? Fig. 4. What does “an average of 2 replicates per sample were used” mean? How many mouse samples per group? Fig. 8, 11, and text – the term “reversed” is used inappropriately; unless the animals received SAA first and induced the outcome then received HDL the HDL can at best be stated as preventative. Fig. 10. Why is SAA stated at 100% and not the control group set as the reference group? Fig. 11. What is the NEG control?

Author Response

Point-by-point response

The following text contains exerts from the revised manuscript shown in color highlight.  Green highlight refers to text that from the original submission (retained in this revised version) and yellow highlighted text refers to additional text requested by the reviewer.

Reviewer 1

The stated goal of the manuscript is to determine the effects of HDL administration on SAA-elicited inflammatory changes in mice. The stated goal of the manuscript is to determine the effects of HDL administration on SAA-elicited inflammatory changes in mice. While there are a number of strengths including the plethora of data, one of the major concerns is that the study is not clearly hypothesis driven, and the reasons for the various measurements are not clear. The authors clearly state that there is overlap between their prior publication for groups 1-3, and that the addition of group 4 is novel, but there are a number of concerns that must be clarified before this study is acceptable.

What is the hypothesis of the overall study, and how does it include both vascular and renal tissues? What is the rationale for addressing HDL impact on SAA? It is danced around, but really not clearly addressed. The rationale for the various tissues and factors measured need to be clearly addressed. At this point it seems that they just measured a whole range of things, but the link is vague at best.

We have clarified the issues raised by the assessor by adding relevant text to the Introduction that now highlights the hypothesis and rationale for measuring the biomarkers we have included in the original manuscript. Please refer to:

Original text on page 3 paragraph 2: “However, the potential for SAA to instigate endothelial [11-13] and renal dysfunction has been established [14]”; this highlights the linkage between changes to vasculature and kidneys, which is the focus of the work conducted in this study. Additional explanatory text on page 3 paragraph 2: “that can initiate the earliest stages of vascular disease and accelerate rates of CVD and renal disorders.” Original text on page 4 paragraph 1: “SAA enrichment can impair anti-inflammatory properties of HDL as shown in patients with diabetic nephropathy [19] and may interfere with HDL’s modulation of pro-atherogenic modifications to low-density lipoprotein (LDL), endothelial cell adhesion molecules (ICAM/VCAM) …….. HDL also inhibits SAA-mediated reactive oxygen species generation and Nod-like receptor protein 3 (NLRP3) inflammasome activation [22].” ; this highlights the involvement of adhesion molecules e.g., VCAM, reactive oxygen species that cause oxidative damage and inflammatory pathways – hence we investigated these pathways using specific biomarkers. Additional explanatory text on Page 4 end of last paragraph: “In this context the current study will test the hypothesis that administration of native human HDL will inhibit recombinant SAAs propensity to elicit pro-inflammatory activity, stimulate oxidative damage and enhance aortic and renal dysfunction in Apo E-/- mice mouse model of atherosclerosis.” Page 17, paragraph 1: “The kidneys are a highly vascularised organ and prone to altered blood flow typically elicited by changes in vascular tone. As SAA promotes vascular dysfunction and altered vascular NO bio-activity (as demonstrated above), it follows that organs such as the kidney may be impacted by SAA stimulation.” Discussion, page 19, bottom of paragraph 2 that now reads:

“Whether, HDL that is known to bind SAA can prevent this SAA-stimulated acceleration of atherosclerosis in ApE-/- mice is the focus of this study.”

Peprotech SAA has been generally disregarded as it is not physiological but rather a recombinant SAA that has features of SAA1 and SAA2, and activities that were not necessarily reproduced by SAA1 or SAA2. The authors need to justify their selection of recombinant SAA for these studies as opposed to studying isolated murine or human SAA.

We agree with the reviewer that the SAA used here is not a physiological representation of biological SAA.  The proposal to use isolated human of mouse SAA has some merit but to clarify the use of this synthetic SAA we have added the following to the experimental section on Page 5, paragraph 1 to highlight the use of this recombinant form of SAA:

Notably, recombinant SAA is not regarded as a physiological analogue of hepatic SAA as it represents a hybrid molecule of isoforms SAA1 and SAA2. While laborious isolation and purification to remove LPS contamination of isolated human or murine SAA is feasible, the difficulties in acquiring suitably pure quantities for use in the planned in vivo studies here ruled out this mode of isolation and despite limitations, the recombinant source was used as an experimental model for SAA protein.

Furthermore, we have previously identified the debate in relation to the use of recombinant SAA in studies with some recent papers providing contradictory arguments for the use of recombinant SAA (See Ref 34 last page of discussion), which on balance do not rule out using this recombinant protein as a model for SAA.

The details re administration of HDL and SAA in group 4 is unclear. Was the SAA administered i.p. after each iv injection of HDL? or did mice receive 2 weeks of HDL then 2 weeks of SAA?

We have now clarified this in the Methodology section on page 7 of the revised manuscript adding the following text to the revised manuscript:

iv) HDL group: Mice designated to receive HDL supplements were pre-injected 100 μL of stock purified human high-density lipoprotein (HDL) (freshly isolated human HDL preparations were diluted in sterile PBS to yield a stock concentration of 1 mg HDL protein/mL; total 100 μg HDL protein/per kg mouse) every third day via tail vein injection for 14 days prior to treatment with SAA (as described above in group (iii)). In summary, mice received HDL for two weeks prior to the administration of SAA for the ensuing 2 weeks.

The authors speculate that the HDL:SAA ratio may be meaningful but there is no evidence that they determined the “fate” of the injected SAA. Did it associate with HDL? Did it elevate systemic SAA in the mice (who also express endogenous SAA)? What was the half-life of the injected SAA? While the LPS group did not appear to differ from the control group the injection of LPS would be expected to induce an inflammatory response in the mice including the induction of endogenous SAA. How high did the SAA go in these mice, and how long did it stay elevated? This should be compared to group 3 (SAA) and group 4 (HDL+SAA).

Note, we have shown previously that the level of contaminating LPS detected in the recombinant SAA is unable to elicit an inflammatory response from isolated human PBMC, a cell type that is highly sensitive to LPS (refer to revised text on page 5, paragraph 1 “Thus, the recombinant SAA employed in this study contained low-level LPS contamination, which has been demonstrated to be relatively inert and unable to stimulate pathophysiologically meaningful pro-inflammatory/pro-coagulant responses in ex vivo studies with isolated human peripheral blood monocytes; a cell type known to be highly responsive to LPS stimulation [35].). 

Accordingly, we anticipate that the LPS control used here would not show excessive inflammation and this is the case as demonstrated here in the various biomarkers of inflammation employed.  Other issues raised by the reviewer in regard to the fate of SAA remain outside the scope of the present study and have been added as a limitation in the discussion (page 24, paragraph 2, see “Study Limitations”) with revised text as follows:

Thus, while we speculate the binding of added SAA by co-administered HDL is a mechanism that explains the protective action of HDL, we have not provided evidence that this occurs in vivo.  To address this, it may be possible to label SAA and track the labelled protein and test whether it can bind detected in HDL isolated from the circulation.  However, this remains outside the scope of the present study and although warranted to confirm the proposed mechanism this proposed mechanism for HDL interaction with SAA remains to be established and our proposal that SAA binds to HSL is based solely on literature evidence.  Furthermore, whether administered recombinant SAA stimulates endogenous SAA production and the half-life of the added SAA in the mouse circulation has not been explored and this is an additional limitation of the current study.

There is no control group for HDL injection. It is possible that HDL alone could have these effects, or that HDL alone could have greater effects that were attenuated by HDL+SAA

The hypothesis that we have included at the request of the reviewer now clearly states that we are testing whether HDL will inhibit recombinant SAAs propensity to elicit pro-inflammatory activity, stimulate oxidative damage and enhance aortic and renal dysfunction in Apo E-/- mice mouse model of atherosclerosis. 

While a further control group comprising HDL alone may attenuate the extent of lesion in the control (PBS) group (attributable to the known anti-atherogenic action of this lipoprotein) it is unclear what interpretation(s) could be added in light of this outcome relative to the main goal to show that intervention with HDL had an effect on the enhanced rate of atherosclerosis induced by supplemented SAA.

Was group 4 enrolled at the same time as groups 1-3, or were groups 1-3 a historical control? This is critical, as various studies have shown that unmeasured factors can impact outcomes – such as time of year, variations in the diet, etc

All mouse cohorts were studied simultaneously, and the data comparing control, LPS and SAA groups has been published recently (Ref 34) however, the impact of HDL on the pro-atherogenic and pro-inflammatory activity of SAA was not published in the same paper and instead we are now including this novel data with the data obtained contemporaneously from vehicle and LPS controls as well as SAA-stimulated mouse cohorts in this current manuscript.

We have added a statement on page 8, paragraph 1 of the revised manuscript to clarify the relationship of this study to the recent publication in Frontiers Immunology (Ref 34):

“Note, the suite of organs harvested from mice assigned to the vehicle and LPS control and SAA groups were recently used to study the impact of SAA stimulation on the vascular endothelium and kidneys and these data have been reported [34]. However, the role for HDL in affecting SAA-mediated inflammation determined in parallel in the same matching ApoE-/- mouse cohort at the same time in parallel was not reported, and this is now the focus of this current study.  Thus, while the impact of SAA stimulation on vascular inflammation and renal dysfunction has been reported in Apo E-/- mice, determination of accelerated aortic lesion formation and altered inflammatory status of cardiac tissues, and the impact of HDL supplementation on these SAA-stimulated changes to aorta and kidneys are novel and not available elsewhere in the literature to the best knowledge of the authors.

There is no investigation or discussion of how HDL may attenuate SAA effects.

The Reviewer correctly points out that the precise mechanism of action has not been determined experimentally in this study and we have provided observational data to support the notion that HDL somehow impacts on SAA pro-atherogenic activity.

To address this deficiency, we have added text to the limitations section in the revised manuscript – specifically:

“Thus, while we speculate the binding of added SAA by co-administered HDL is a mechanism that explains the protective action of HDL we have not provided evidence that this occurs in vivo.  To address this, it may be possible to label SAA and track the labelled protein and test whether it can bind to HDL isolated from the circulation.  However, this remains outside the scope of the present study and although warranted to confirm the proposed mechanism this proposed mechanism for HDL interaction with SAA remains to be established and our proposal that SAA binds to HSL is based solely on literature evidence.”

Minor points:

The figure legends need to be consistent – the number per group is not always stated and needs to be.

Where relevant, we have addressed this issue in the legends to all figures.

Fig 1 – atherosclerosis data is usually not normally distributed and should not be presented as mean +/- SD, but generally as a dot plot.

This approach has been adopted particularly in human studies where data from individuals are readily separated and shown in the distribution (dot blot) format described by the Reviewer.  However, it is not uncommon that animal studies present data as grouped from single cages to maintain consistency with environmental factors and data presented as mean data without distribution (refer to Bräsen et al Atherosclerosis, 2002, 163; 249-259; Witting et al FASEB J, 1999, 13;667-675) as the case here.

Fig 1 – the presentation of aortic root atherosclerosis as a % lesion area is atypical. What was used as the denominator (total area) and did that differ between mice – e.g., from variable perfusion pressure or tissue structure/artefact in section preparation. A more standard measure is to cite atherosclerotic area in um2.

We have added text to the methodology section (page 9, paragraph 3 in the revised manuscript) to describe the process of normalising lesion to the corresponding lumen measured at the aortic root in the same section.  The following text has been included:

In the case of the aortic root, care was taken to ensure that the image captured sections contained valve leaflets at the same anatomical level in the hearts from the different treatment groups as we have published previously [38].  This approach yielded lesion area as a relative % of the total luminal area at the same anatomical region of the aortic root as defined by using Image J to map around the internal lumen that defined the aortic root in the section being investigated.

We also note that we defined the fractional or % lesion in the corresponding results section (page 14, paragraph 1) that read:

Determinations of the fractional lesion area relative to the lumen of the aortic root revealed a significant increase in the % lesion size in Apo E-/- mice treated with SAA alone

….and we now add the definition of % Lesion to the figure legend to Figure 1 that reads…..” relative % lesion size (determined as lesion area/corresponding luminal area at the aortic root calculated using Image J software (v1.42, NIH, USA).”

The authors should state if they adhered to the scientific statement on animal atherosclerosis studies by Daugherty et al (ATVB 2017).

In terms of the reference to the Alan Daugherty statement paper:

We note that while it is common practice to use genetically modified mice as an experimental model of atherosclerosis, marked variations in outcomes between laboratories and a general lack of clinical translation of outcomes from mouse models to humans can complicate the available literature and future studies derived from this literature base.  We acknowledge the statement of practice as set out in the “Recommendation on Design, Execution, and Reporting of Animal Atherosclerosis Studies: A Scientific Statement From the American Heart Association” and identify the following in relation to the study reported here:

We have used a common genetically modified mouse model to assess atherosclerosis and accept that the shortcomings of this model, as identified in the statement, are a general limitation of this experimental model. Further validating experiments using non-rodent based animal models (such as rabbits, pigs and non-human primates) are absolutely required before the conclusions drawn from this study can be translated to human conditions where eleavyed SAA levels may impact on vascular and renal function. In the experimental design, we have refrained from using a high fat diet to accelerate atherosclerosis as the hypothesis being tested is that SAA itself plays a role in promoting proatherogenic factors that accelerate atherosclerosis and so a high fat diet would interfere with this assessment. We have used a reputable supplier of ApoE-/- mice in Australia (Animal Resources Centre (Perth, Western Australia)) that supply a range of mice for research purposes and mice were contained in the same environment within the animal facility with the access to the same chow and water supply. Mice (8 weeks old) were transported to a local site for husbandry, allowed to acclimate and then were randomly divided into four groups without internal bias. Data obtained using (i) analytical and (ii) imaging techniques reported in this study were repeated using the same tissues at the same time for all of the treatment cohorts to gain a valid and rigorous comparison between vehicle, LPS, SAA and HDL-intervention cohorts. We have reported how often a given experiment was repeated to substantiate the outcome, and this was established using the nominated statistical tests with appropriate corrections.

This statement and the reference cited by the Reviewer has been added to the limitations section in the Discussion of the revised manuscript.

Fig 2. Is there any staining in the control group? What is the arrow indicating?

The Reviewer refers to the description in the results section on page 14 (bottom) in the revised manuscript.  We have clarified that staining appears as a “dark granular appearance at the cell surface” on the endothelium and note that there was virtually no VCAM-1+ staining in the vehicle control and LPS samples:

“As we described previously [34], discernibly higher VCAM-1+ (dark granular accumulation at the cell surface) immunostaining was observed in thoracic aortae from the SAA group (Figure 2 panels C/G) than corresponding aortic tissue from the vehicle control (Figure 2 panels A/E) and LPS groups (Figure 2 panels B/F) that showed virtually no staining of the endothelium.”

The arrow refers specifically to the luminal surface of the endothelium in each image.  This is now clearly identified in the corresponding legend to Figure 2 that reads:

“Black arrows indicate the endothelial surface on the inner lumen of the artery in panels A-D. Dark stained immuno-reactive VCAM-1 on the vascular endothelium is evident in Panel C whereas it is markedly lower or absent in Panels A, B and D. The panels in the right column show a high magnified view of corresponding boxed regions of endothelial with VCAM-1+ immuno-reactivity seen as a granular deposit on the endothelium.”

Fig. 4. What does “an average of 2 replicates per sample were used” mean? How many mouse samples per group?

We have clarified this in the legend to Figure 4 in the revised manuscript as requested by the Reviewer and now include the following explanation:

“Each aortic homogenate obtained from control (n = 8), LPS (n = 8) and SAA (n = 8), HDL/SAA (n=8) mice was sampled twice, and the output averaged from these 2 replicate measurements of oxidized lipid and then expressed as mean ± SD for each mouse cohort.”

Fig. 8, 11, and text – the term “reversed” is used inappropriately; unless the animals received SAA first and induced the outcome then received HDL the HDL can at best be stated as preventative.

As requested, we have removed reverse throughout the text and replaced the word with prevent or inhibit in the revised manuscript.

Fig. 10. Why is SAA stated at 100% and not the control group set as the reference group?

As requested, we have re-expressed the % change in each group using the control outcome as the reference set point for each cytokine/chemokine studied in the revised manuscript.

Fig. 11. What is the NEG control?

The ideal approach in immunofluorescent studies is to run a negative control which does not have a primary antibody but only the secondary antibody. This ensures that the signal we get in the test samples incubated with primary and secondary antibodies is a real signal coming from the complex of primary antibody and the target protein of interest and not a background or artefact.

We have now defined the negative or NEG control as the use of secondary antibody alone.  This information is now in the legend for Figure 11 in the revised manuscript and in the corresponding experimental section (Page 13, paragraph 1) that now reads:

“During each run negative control slides (corresponding to treatment of the tissue with secondary antibody alone) were assessed in parallel to ensure the efficacy and reliability of antibodies and validate the immunostaining technique”.

Reviewer 2 Report

In the present study, the authors investigate the effect of co-supplementing HDL on SAA-mediated changes to vascular and renal function in apolipoprotein E-deficient (ApoE -/- ) mice in the absence of a high-fat diet. They showed that SAA treatment increases adhesion molecule and promote endothelial dysfunction. Moreover, they observed an increase of urinary protein, kidney injury molecule (KIM)-1 and renal tissue cytokine/chemokine levels concomitantly with an increase of P38 activation, confirming that SAA elicited a pro-inflammatory phenotype in the kidney. Of note, supplementation with HDL significantly reduces kidney inflammatory status. They conclude that supplementation with HDL reduces SAA-mediated endothelial and renal dysfunction in an atherosclerosis prone mouse model

The manuscript is interesting, but several issues should be addressed before being reconsidered.

The present manuscript is based on authors previous publication (Frontiers in Immunology doi: 10.3389/fimmu.2019.00380) in which animals have been subjected to Normal Chow Diet. Please, discuss more precisely the differences between the two studies and what it adds to the literature.

Laboratory parameters of all mice groups, including creatinine levels and ions, must be reported.

Figure 2 is difficult to interpret. The authors indicate with the arrow the immunoreactivity to VCAM1. Whereas, the insets highlight other sites. It is not clear.

Figure 3A. Please report the marker on the Y-axis.

What are the arginase activity in hearts and renal tissues?

What is the status of the glomerular area? The authors should evaluate % of Bowman’s space to assess the protective action of HDL.

What is the status of ROS in all animal groups? The Authors should assess 8-isoprostane or TMAO levels to evaluate if its antioxidant properties could mediate the possible beneficial effect of HDL.

The authors should evaluate by immunohistochemistry the level of monocytes/macrophage infiltration both in heart and aorta.

Author Response

Point-by-point response

The following text contains exerts from the revised manuscript shown in color highlight.  Green highlight refers to text that from the original submission (retained in this revised version) and yellow highlighted text refers to additional text requested by the reviewer.

Reviewer 2

In the present study, the authors investigate the effect of co-supplementing HDL on SAA-mediated changes to vascular and renal function in apolipoprotein E-deficient (ApoE -/- ) mice in the absence of a high-fat diet. They showed that SAA treatment increases adhesion molecule and promote endothelial dysfunction. Moreover, they observed an increase of urinary protein, kidney injury molecule (KIM)-1 and renal tissue cytokine/chemokine levels concomitantly with an increase of P38 activation, confirming that SAA elicited a pro-inflammatory phenotype in the kidney. Of note, supplementation with HDL significantly reduces kidney inflammatory status. They conclude that supplementation with HDL reduces SAA-mediated endothelial and renal dysfunction in an atherosclerosis prone mouse model

The manuscript is interesting, but several issues should be addressed before being reconsidered.

The present manuscript is based on authors previous publication (Frontiers in Immunology doi: 10.3389/fimmu.2019.00380) in which animals have been subjected to Normal Chow Diet. Please, discuss more precisely the differences between the two studies and what it adds to the literature.

All mouse cohorts were studied simultaneously, and the data comparing control, LPS and SAA groups has been published recently (Ref 34) however, the impact of HDL on the pro-atherogenic and pro-inflammatory activity of SAA was not published in the same paper and instead we are including this novel data with the data from control, LPS and SAA groups in this current manuscript.

We have added a statement on page 8, paragraph 1 of the revised manuscript to clarify the relationship of this study to the recent publication in Frontiers Immunology (Ref 34):

“Note, the suite of organs harvested from mice assigned to the vehicle and LPS control and SAA groups were recently used to study the impact of SAA stimulation on the vascular endothelium and kidneys and these data have been reported [34]. However, the role for HDL in affecting SAA-mediated inflammation determined in parallel in the same matching ApoE-/- mouse cohort at the same time in parallel was not reported, and this is now the focus of this current study.  Thus, while the impact of SAA stimulation on vascular inflammation and renal dysfunction has been reported in Apo E-/- mice, determination of accelerated aortic lesion formation and altered inflammatory status of cardiac tissues, and the impact of HDL supplementation on these SAA-stimulated changes to aorta and kidneys are novel and not available elsewhere in the literature to the best knowledge of the authors.

Laboratory parameters of all mice groups, including creatinine levels and ions, must be reported.

We have not assessed the parameters indicated in the plasma/urine that was harvested.  Presently, we have no urine samples remaining and an incomplete set of plasma samples (this was used to generate extensive (native and oxidized) plasma lipid and antioxidant data – see Ref 34), so the requested additional analyses cannot be determined unless the study is completely repeated.

We do note however, that we have monitored several urinary markers (total protein and KIM-1) as a surrogate for renal dysfunction and also determined several renal tissue parameters (related to inflammatory state) that consistently an strongly correlate experimental renal dysfunction.

Figure 2 is difficult to interpret. The authors indicate with the arrow the immunoreactivity to VCAM1. Whereas, the insets highlight other sites. It is not clear.

The Reviewer refers to the description in the results section on page 14 (bottom) in the revised manuscript.  We have clarified that staining appears as a “dark granular cell surface” on the endothelium and note that there was virtually no VCAM-1+ staining in the vehicle control and LPS samples:

“As we described previously [34], discernibly higher VCAM-1+ (dark granular surface staining) immunostaining was observed in thoracic aortae from the SAA group (Figure 2 panels C/G) than corresponding aortic tissue from the vehicle control (Figure 2 panels A/E) and LPS groups (Figure 2 panels B/F) that showed virtually no VCAM-1+ staining of the endothelium.”

The arrow refers specifically to the luminal surface of the endothelium in each image.  This is now clearly identified in the corresponding legend to Figure 2 that reads:

“Black arrows indicate the endothelial surface on the inner lumen of the artery in panels A-D. Dark stained immuno-reactive VCAM-1 on the vascular endothelium is evident in Panel C whereas it is markedly lower or absent in Panels A, B and D. The panels in the right column show a high magnified view of corresponding boxed regions of endothelial with VCAM-1+ immuno-reactivity seen as a granular deposit on the endothelium.”

Figure 3A. Please report the marker on the Y-axis.

As requested, we have added the legend “Fold-change to VCAM-1 mRNA expression” to Figure 3A in the revised manuscript.

What are the arginase activity in hearts and renal tissues?

This parameter was not consistently measured in these tissues although aortic arginase activity has been reported in our earlier publication (Ref 34) - we did not complete the corresponding parallel analyses in the samples from the HDL/SAA co-supplemented cohort and so we are unable to make a valid comparison with this parameter.

What is the status of the glomerular area? The authors should evaluate % of Bowman’s space to assess the protective action of HDL.

As requested, we have added the additional data in the revised manuscript.  The corresponding description of this additional data reads (Page 17):

“In addition to measurement of this biochemical measure of renal tubule damage we also assessed changes to renal glomeruli in the different treatment groups (Figure 8B).  The data indicate that administered LPS marginally decreased Bowman’s space (decreased 10% vs vehicle control, p<0.05), whereas Bowman’s space was more markedly decreased (decreased 15% vs vehicle control, p<0.05) in the presence of SAA.  Notably, HDL co-administration prevented this SAA-stimulated decrease in Bowman’s space (decreased 3% vs vehicle control, p<0.05) with glomeruli appearing less condensed and more similar to corresponding glomeruli in the control.”

This outcome has been included in the abstract in the revised submission.

What is the status of ROS in all animal groups? The Authors should assess 8-isoprostane or TMAO levels to evaluate if its antioxidant properties could mediate the possible beneficial effect of HDL.

We have already measured two independent parameters of ROS-mediated damage that are known to result from free radical-mediated damage to lipid (F2-isoprostane) and inflammatory pathways (the latter yielding hypochlorous acid as oxidant, which promotes the protein damage, assessed here as 3-Cl-Tyr).  We selected these biomarkers as they are most relevant to SAA-mediated damage to cells and tissues as we have previously identified (See Ref 13 and 34 in the revised manuscript).  Therefore, we have not followed the suggestion to add additional markers of damage that have not been linked to SAA-mediated damage.

The authors should evaluate by immunohistochemistry the level of monocytes/macrophage infiltration both in heart and aorta.

It is not clear whether SAA impacts on monocyte density and maturation to macrophages and so we have not measured these parameters in the lesions that have been identified in the aortic root of these mice.

However, as indicted in the response to Reviewer 1, point 1 (shown below), we have selected the raft of biomarkers used in this study that were most relevant to SAA-mediated inflammation and oxidative tissue damage. The following new text has been revised in the original manuscript: Please refer to:

Original text on page 3 paragraph 2: “However, the potential for SAA to instigate endothelial [11-13] and renal dysfunction has been established [14]”; this highlights the linkage between changes to vasculature and kidneys, which is the focus of the work conducted in this study. Additional explanatory text on page 3 paragraph 2: “that can initiate the earliest stages of vascular disease and accelerate rates of CVD and renal disorders.” Original text on page 4 paragraph 1: “SAA enrichment can impair anti-inflammatory properties of HDL as shown in patients with diabetic nephropathy [19] and may interfere with HDL’s modulation of pro-atherogenic modifications to low-density lipoprotein (LDL), endothelial cell adhesion molecules (ICAM/VCAM) …….. HDL also inhibits SAA-mediated reactive oxygen species generation and Nod-like receptor protein 3 (NLRP3) inflammasome activation [22].” ; this highlights the involvement of adhesion molecules e.g., VCAM, reactive oxygen species that cause oxidative damage and inflammatory pathways – hence we investigated these pathways using specific biomarkers. Additional explanatory text on Page 4 end of last paragraph: “In this context the current study will test the hypothesis that administration of native human HDL will inhibit recombinant SAAs propensity to elicit pro-inflammatory activity, stimulate oxidative damage and enhance aortic and renal dysfunction in Apo E-/- mice mouse model of atherosclerosis.”

Reviewer 3 Report

This study describes that HDL inhibits SAA-induced vascular and renal dysfunctions in Apolipoprotein E-deficient mice. Overall the study presented is sound, however, the following points should be considered for improving the flow of the study:

Introduction

– please improve the cohesiveness of the last paragraph in the introduction to increase the clarity of why the study was conducted.

– please provide an explanation for why the 4-, 8- or 18-week time-points were chosen in the study.

Methods

– section 2.2.2. please include the ethics approval number with which the study was conducted.

– section 2.2.3. please include the ethics approval number with which the study was conducted.

Results

– Fig 1. Please make the location of the aortic lesions clearer, the authors may want to replicate the representation utilised in fig. 2

– Fig 2. For panels A, B and D, please clarify where the VCAM-1 staining is localised, as it is not as clear as the data shown in panel C.

– Fig 6. Please indicate why selected inflammatory cytokines were measured and not a broader range.

Author Response

Point-by-point response

The following text contains exerts from the revised manuscript shown in color highlight.  Green highlight refers to text that from the original submission (retained in this revised version) and yellow highlighted text refers to additional text requested by the reviewer.

Reviewer 3

This study describes that HDL inhibits SAA-induced vascular and renal dysfunctions in Apolipoprotein E-deficient mice. Overall the study presented is sound, however, the following points should be considered for improving the flow of the study:

Introduction

– please improve the cohesiveness of the last paragraph in the introduction to increase the clarity of why the study was conducted.

As indicated in the response to Reviewer 1, major point 1 (also pasted below) – we have now refined the introduction to indicate the hypothesis to be tested and refined the text to better identify the reasons why SAA pro-inflammatory activity on aortic vessels and renal tissues was the focus of study.

Please refer to:

Original text on page 3 paragraph 2: “However, the potential for SAA to instigate endothelial [11-13] and renal dysfunction has been established [14]”; this highlights the linkage between changes to vasculature and kidneys, which is the focus of the work conducted in this study. Additional explanatory text on page 3 paragraph 2: “that can initiate the earliest stages of vascular disease and accelerate rates of CVD and renal disorders.” Original text on page 4 paragraph 1: “SAA enrichment can impair anti-inflammatory properties of HDL as shown in patients with diabetic nephropathy [19] and may interfere with HDL’s modulation of pro-atherogenic modifications to low-density lipoprotein (LDL), endothelial cell adhesion molecules (ICAM/VCAM) …….. HDL also inhibits SAA-mediated reactive oxygen species generation and Nod-like receptor protein 3 (NLRP3) inflammasome activation [22].” ; this highlights the involvement of adhesion molecules e.g., VCAM, reactive oxygen species that cause oxidative damage and inflammatory pathways – hence we investigated these pathways using specific biomarkers. Additional explanatory text on Page 4 end of last paragraph: “In this context the current study will test the hypothesis that administration of native human HDL will inhibit recombinant SAAs propensity to elicit pro-inflammatory activity, stimulate oxidative damage and enhance aortic and renal dysfunction in Apo E-/- mice mouse model of atherosclerosis.”

– please provide an explanation for why the 4-, 8- or 18-week time-points were chosen in the study.

Additional text has been added to the methods section indicated by the Reviewer to justify the use of the time points selected:  Refer to page 7 (bottom) of the revised manuscript that now reads:

“Animals were sacrificed at two time points after commencement of treatment. These time points were selected based on screening the extent of SAA-stimulated inflammation detected previously in aorta and kidney [34].

4 weeks after commencement of SAA treatment (referred to as the young group representing an acute model and judged to be a suitable time for assessing early general pro-inflammatory changes in aortic and renal tissues) and 18 weeks after commencement of SAA treatment (referred to as the old group representing a chronic model suitable for assessing marked phenotypic changes to the vasculature including the development of quantifiable aortic lesion)”

Methods

– section 2.2.2. please include the ethics approval number with which the study was conducted.

This information is now added to page 6 of the revised manuscript (AEC approval #2013/041)

– section 2.2.3. please include the ethics approval number with which the study was conducted.

This information is now added to page 6 of the revised manuscript (AEC approval #2013/041)

Results

– Fig 1. Please make the location of the aortic lesions clearer, the authors may want to replicate the representation utilised in fig. 2

As suggested by the reviewer we have highlighted and labelled the lesion area in each section in the revised Figure 1 for clarity.

We have also added some addition text to describe the complexity of the lesions that are formed especially in the presence of SAA that has clearly promoted an accelerated lesion formation with evidence of a complex necrotic core with cholesterol crystals, not evident in the vehicle, LPS controls or the aortic root from mice supplemented HDL/SAA.

This now includes the following text in the revised manuscript (Page 14, paragraph 1):

Furthermore, close inspection of the aortic lesion in mice stimulated with SAA showed the presence of complex late stage lesions typified by the formation of a necrotic core containing cholesterol clefts.  By contrast, mice pre-treated with human HDL prior to SAA stimulation showed significantly lower % lesion (HDL+SAA group) than mice supplemented SAA alone (compare Figures 1C with D and refer to E) with a general absence of a highly developed necrotic core.

– Fig 2. For panels A, B and D, please clarify where the VCAM-1 staining is localised, as it is not as clear as the data shown in panel C.

The Reviewer refers to the description in the results section on page 14 (bottom) in the revised manuscript.  We have clarified that staining appears as a “dark granular cell surface” on the endothelium and note that there was virtually no VCAM-1+ staining in the vehicle control and LPS samples:

“As we described previously [34], discernibly higher VCAM-1+ (dark granular cell surface staining) immunostaining was observed in thoracic aortae from the SAA group (Figure 2 panels C/G) than corresponding aortic tissue from the vehicle control (Figure 2 panels A/E) and LPS groups (Figure 2 panels B/F) that showed virtually no staining of the endothelium.”

The arrow refers specifically to the luminal surface of the endothelium in each image.  This is now clearly identified in the corresponding legend to Figure 2 that reads:

“Black arrows indicate the endothelial surface on the inner lumen of the artery in panels A-D. Dark stained immuno-reactive VCAM-1 on the vascular endothelium is evident in Panel C whereas it is markedly lower or absent in Panels A, B and D. The panels in the right column show a high magnified view of corresponding boxed regions of endothelial with VCAM-1+ immuno-reactivity seen as a granular deposit on the endothelium.”

– Fig 6. Please indicate why selected inflammatory cytokines were measured and not a broader range.

The Introduction has been focussed to identify the biological activity of SAA and hence the raft of biomarkers was refined to reflect measurement of this bioactivity.  The chemokines and cytokines studied here were judged to be relevant to the putative bioactivity for SAA and were collected in a single commercial multiplex kit that allowed simultaneous assessment of all the analytes measured.  For this reason, we were limited in the choice of chemokines and cytokines as this was dictated by the mouse multiplex kit (this was not a custom designed kit rather an off the shelf predefined mouse chemokines and cytokines array).  A statement has been added to the relevant methods section to identify this as a reason for the selected assessment of (refer to page 10, paragraph 2) that reads…

“This commercial kit was selected as the raft of chemokines and cytokines included in the kit were judged to be relevant to the putative bioactivity for SAA and were collected in a single multiplex array that allowed simultaneous assessment of all the analytes measured.”

Round 2

Reviewer 2 Report

I have no other comments to authors.